

# Improved Prediction of Dimethyl Sulfide (DMS) Distributions in the NE Subarctic Pacific using Machine Learning Algorithms

Brandon J. McNabb[1] & Philippe D. Tortell[1,2]

[1]Department of Earth, Ocean and Atmospheric Sciences, University of British Columbia, Vancouver, BC V6T 1Z4, Canada

[2]Department of Botany, University of British Columbia, Vancouver, BC V6T 1Z4, Canada

*Correspondence to*: Brandon J. McNabb (bmcnabb@eoas.ubc.ca)

**Abstract.** Dimethyl sulfide (DMS) is a volatile biogenic gas with the potential to influence regional climate as a source of atmospheric aerosols and cloud condensation nuclei (CCN). The complexity of the oceanic DMS cycle presents a challenge in accurately predicting sea-surface concentrations and sea-air fluxes of this gas. In this study, we applied machine learning methods to model the distribution of DMS in the NE Subarctic Pacific (NESAP), a global DMS hot-spot. Using nearly two decades of ship-based DMS observations, combined with satellite-derived oceanographic data, we constructed ensembles of 1000 machine-learning models using two techniques, random forest regression (RFR) and artificial neural networks (ANN). Our models dramatically improve upon existing statistical DMS models, capturing up to 62% of observed DMS variability in the NESAP and demonstrating notable regional patterns that are associated with mesoscale oceanographic variability. In particular, our results indicate a strong coherence between DMS concentrations, sea surface nitrate (SSN) concentrations, photosynthetically active radiation (PAR) and sea surface height anomalies (SSHA), suggesting that NESAP DMS cycling is primarily influenced by heterogenous nutrient availability, light-dependent processes and physical mixing. Based on our model output, we derive summertime, sea-air flux estimates of 1.16±1.22 Tg S in the NESAP. Our work demonstrates a new approach to capturing spatial and temporal patterns in DMS variability, which is likely applicable to other oceanic regions.

# 1 Introduction

Dimethyl sulfide (DMS), a volatile biogenic gas, is an important component of the marine sulfur cycle. This molecule is an important substrate for specific methylotrophic bacteria (Vila-Costa et al., 2006; Lidbury et al., 2016; Green et al., 2011; Hatton et al., 2012), with a recognized importance to marine microbial metabolism (Vila-Costa et al., 2006) and food web interactions (Nevitt, 2008). Moreover, DMS constitutes the largest fraction of bulk non-sea salt (NSS) sulfate emissions to the atmosphere (Bates et al., 1992; Ksionzek et al., 2016), where it is rapidly oxidized to form aerosols that act as cloud condensation nuclei (CCN; Charlson et al., 1987; Hegg et al., 1991; Korhonen et al., 2008), potentially influencing regional albedo and climate (Charlson et al., 1987; Ayers and Cainey, 2007). Given the ecological roles of DMS and its potential influence on global climate, substantial research has focused on characterizing the dynamics of this compound in seawater. This work has revealed considerable complexity in the oceanic DMS cycle, which has limited the development of simple predictive algorithms describing its spatial and temporal variability.

Oceanic DMS production and loss are tightly linked with the biological cycling of the related metabolites dimethyl sulfoniopropionate (DMSP) and dimethyl sulfoxide (DMSO). DMS is believed to be primarily derived

from the cleavage of DMSP (Kiene and Linn, 2000), but it can also be cycled through biological DMSO reduction (Spiese et al., 2009) and oxidation (Lidbury et al., 2016), and abiotically by light-dependent reactions (del Valle et al., 2007; Royer et al., 2016). DMS cycling is influenced by a suite of environmental and ecological factors, including release from phytoplankton cells into the dissolved pool via grazing (Dacey and Wakeham, 1986), viral lysis (Malin et al., 1998), or exudation. Oxidative stress generated by other variables such as temperature (Kirst et al., 1991), salinity (Dickson and Kirst, 1987), UV radiation (Kinsey et al., 2016), and nutrient limitation (Bucciarelli et al., 2013; Spiese & Tatarkov, 2014) may also enhance the cycling of DMSP and DMSO, which may regulate DMS concentrations through cascading oxidative pathways (Sunda et al., 2002). Finally, variability in surface wind fields can modulate the rates of DMS sea-air exchange, providing a significant source of heterogeneity in surface water DMS concentrations (Royer et al., 2016). These examples illustrate the complex non-linearity of the oceanic DMS cycle.

Over the past two decades, a number of approaches have been developed to model DMS distributions at both global (Bock et al., 2021; Galí et al., 2018; Simó and Dachs, 2002; Vallina and Simó, 2007) and regional (Watanabe et al., 2007) scales. These models have been largely based on linear regression techniques estimating DMS concentrations using one or two predictors. To date, these studies have focused on a number of variables, including ratio of chlorophyll a (Chl-a) to mixed layer depth (MLD) (Simó and Dachs, 2002), sea surface temperature (SST) and nitrate (SSN) (Watanabe et al., 2007), solar radiation dose (SRD) (Vallina and Simó, 2007), photosynthetically active radiation (PAR) and modelled DMSP concentrations (Galí et al., 2018). Some of these models have demonstrated reasonably good performance at global scales, but their predictive power is generally diminished at regional scales (Herr et al., 2019), failing to accurately resolve important smaller-scale features (Belviso et al., 2003; Nemcek et al., 2008; Royer et al., 2015; Tortell, 2005b).

In recent years, machine-learning algorithms have been increasingly used to derive predictions for non-linear oceanic systems. For example, these methods have been successfully applied to describe the spatial and temporal patterns of global methane flux (Weber et al., 2019), nitrous oxide dynamics (Yang et al., 2020), and carbon export (Roshan and DeVries, 2017). To our knowledge, only two studies have thus far applied machine-learning to describe DMS distributions, with one study focused on the Arctic (Humphries et al., 2012) and the other exploring a global domain (Wang et al., 2020). Despite producing algorithms with reasonable predictive skill, these two studies found limited success in resolving the underlying relationships driving DMS variability. This was partially due to a reliance on indirect sensitivity tests assessing the importance of predictor variables, and also, potentially, from the large-scale averaging applied to the underlying data fields ($1 \times 1°$; $111$ km$^2$). Analyses at higher

spatial resolution may reveal mesoscale (roughly 20-200 km) and sub-mesoscale (roughly 1-20 km) patterns that
would otherwise be obscured, thereby increasing predictive strength.
Machine learning algorithms require large datasets for the training and testing process. Traditionally, DMS
measurements were based on time-consuming ship-board analysis of discrete samples, resulting in sparse data
coverage over much of the oceans. More recently, the development of several automated DMS measurement
systems (Royer et al., 2014; Saltzman et al., 2009; Tortell, 2005a) has provided marine DMS observations at a
significantly higher resolution, yielding greater spatial and temporal data coverage. These new datasets potentially
enable new insights into small-scale and regional patterns in oceanic DMS distributions, as well as the
characterization of oceanic DMS 'hot-spots'. The northeast subarctic Pacific (NESAP) is a region of notably high
DMS concentrations (Lana et al., 2011), with localized DMS accumulation in both highly productive coastal
upwelling regimes, and off-shore, iron-limited waters (Herr et al., 2019; Asher et al., 2017). Several factors have
been proposed to account for the elevated DMS production in the NESAP, including increased primary productivity
driven by nutrient entrainment and upwelling along coastal fronts (Asher et al., 2017), a dominance of high-DMSP
producing prymnesiophytes and dinoflagellates in offshore waters, elevated microbial degradation of DMSP to
DMS (Steiner et al., 2012; Royer et al., 2010), and the stimulation of DMS production in response to oxidative
stress in low iron waters (Sunda et al., 2002; Herr et al., 2020). Although multiple studies have examined empirical
relationships between DMS and various oceanographic factors in the NESAP (Watanabe et al., 2007; Herr et al.,
2019; Asher et al., 2017, 2011), these have all reported low predictive skill based on simple linear correlation
approaches. To date, machine-learning approaches have not been applied to describe DMS distributions specifically
in this region.
Here, we present an approach to modelling summertime NESAP DMS concentrations and sea-air fluxes
using ensemble random forest regression (RFR) and artificial neural network (ANN) machine-learning algorithms.
Our statistical models leverage field observations of DMS collected across the NESAP between 1997 to 2017 to
generate a summertime DMS climatology mapped at a higher spatial resolution than previous efforts (Simó and
Dachs, 2002; Vallina and Simó, 2007; Galí et al., 2018; Watanabe et al., 2007; Humphries et al., 2012; Wang et
al., 2020). This new modelling approach represents a significant improvement over previous methods and predicts
regional DMS distributions that are coherent with underlying patterns of oceanographic variability. Most notably,
the modelled DMS concentrations and sea-air fluxes can be explained, to a large extent, by regional and mesoscale
patterns in nutrient supply and physical mixing dynamics. Based on the output of our models, we present
summertime sea-air flux estimates in close agreement with previous studies (Herr et al., 2019; Lana et al., 2011),
further highlighting the importance of the NESAP as a globally-significant sulfur source to the atmosphere.

## 2 Methods

### 2.1 Data

A combination of data sources was used in training our machine-learning models to build a summertime DMS climatology. For this study, we restricted DMS measurements to the months of June, July and August between 1997 to 2017 in the NESAP (43-60°N, 147-122°W). A total of 26,201 data points were obtained from the NOAA PMEL repository (https://saga.pmel.noaa.gov/dms/; last accessed: February 3, 2021), including measurements derived from purge and trap gas chromatography and membrane inlet mass spectrometry. The DMS data were binned to a monthly resolution, regardless of year, and averaged into 0.25 x 0.25° grid cells.

Predictor data used to build our machine-learning models included the following variables derived from the NASA Aqua MODIS satellite at level L3 monthly 0.042° resolution (R2018.0): sea surface temperature (SST), the ratio of normalized fluorescence line height to chlorophyll a (nFLH:Chl-a), instantaneous and daily observed photosynthetically active radiation (iPAR and PAR, respectively), particulate inorganic carbon (PIC), the absorption of gelbstof and detritus at 433 nm ($a_{cdm}(443)$), and diffuse attenuation coefficients at 490nm ($K_d$). Satellite-based PIC is considered as a proxy for the abundance of coccolithophores and other calcified phytoplankton (Franklin et al., 2010), whereas the $a_{cdm}(443)$ product is considered a proxy for chromophoric dissolved organic matter (CDOM; Nelson & Siegel, 2013), which is thought to be an important photosensitizer of DMS (see Sect. 4.1). For observations prior to 2004, data were from either SeaWiFS (0.083° resolution) or Terra MODIS (0.042° resolution) when SeaWiFS data were unavailable (*e.g.* nFLH and iPAR). As described below, $K_d$ and PIC were later excluded from the final models (see Sect. 2.6), as they didn't improve predictive skill.

The following predictor variables were also used: 6-day averaged sea surface height anomalies (SSHA) derived from the TOPEX/Poseidon satellites at 0.17° resolution; Level L4 ESA Sentinal-3 Copernicus monthly-averaged 0.25° wind speeds; net primary productivity (NPP) from the Vertically-Generalized Production Model (VGPM; Behrenfeld & Falkowski, 1997) at monthly 0.25° resolution; sea surface nitrate from the 2018 World Ocean Atlas at monthly 1° resolution (Garcia et al., 2019); and mixed-layer depth (MLD) and sea surface salinity (SSS) from the MIMOC climatology at 0.5° resolution (Schmidtko et al., 2013). Except for MIMOC data, all predictors were restricted in time to the corresponding years of DMS sampling (1997 to 2017). Net community productivity (NCP) was estimated from the algorithm of Li & Cassar, (2016; using NPP and SST). As with DMS observations, predictor data were interpolated to a 0.25 x 0.25° average monthly resolution using linear radial basis interpolation functions. Interpolation was constrained to the oceanic region by masking out land pixels using ETOPO2 bathymetric (0.033° resolution) binned at 0.25 x 0.25° resolution. We note that each of these data sources

129 are likely to have inherent uncertainties associated with either their collection or processing. Data sources can be

130 found in Table 1.

131

132 **Table 1. Data sources and spatial and temporal resolution of predictor variables used to develop the RFR and ANN algorithms. Data**
133 **processing levels are indicated where relevant. All variables were used as predictors (excluding bathymetry) and post-processed to**
134 **monthly-averaged, 0.25º resolution (see sections 2.1-2.2).**

| Variable | Spatial Resolution (º) | Temporal Resolution | Source | Level |
|---|---|---|---|---|
| Sea Surface Temperature (SST) | 0.042 | 6-Day Average | SeaWiFS/AquaTERRA (1997-2003) or AquaMODIS(2004-2017): https://oceancolor.gsfc.nasa.gov/l3/ | 3 |
| Chlorophyll-Normalized Fluorescence (nFLH:Chl-a) | 0.042 | Monthly | SeaWiFS/AquaTERRA (1997-2003) or AquaMODIS (2004-2017): https://oceancolor.gsfc.nasa.gov/l3/ | 3 |
| Instantaneous Photosynthetically Active Radiation (iPAR) | 0.042 | Monthly | SeaWiFS/AquaTERRA (1997-2003) or AquaMODIS (2004-2017): https://oceancolor.gsfc.nasa.gov/l3/ | 3 |
| Daily Photosynthetically Active Radiation (PAR) | 0.042 | Monthly | SeaWiFS/AquaTERRA (1997-2003) or AquaMODIS (2004-2017): https://oceancolor.gsfc.nasa.gov/l3/ | 3 |
| Particulate Inorganic Carbon (Calcite; PIC) | 0.042 | Monthly | SeaWiFS/AquaTERRA (1997-2003) or AquaMODIS (2004-2017): https://oceancolor.gsfc.nasa.gov/l3/ | 3 |
| Absorption of Gelbstof and Detritus at 433 nm ($a_{cdm}(443)$) | 0.042 | Monthly | SeaWiFS/AquaTERRA (1997-2003) or AquaMODIS (2004-2017): https://oceancolor.gsfc.nasa.gov/l3/ | 3 |
| Diffuse Attenuation Coefficients at 490 nm ($K_d$) | 0.042 | Monthly | SeaWiFS/AquaTERRA (1997-2003) or AquaMODIS (2004-2017): https://oceancolor.gsfc.nasa.gov/l3/ | 3 |
| Sea Surface Height Anomalies (SSHA) | 0.17 | Monthly | TOPEX/Poseidon: https://podaac.jpl.nasa.gov/dataset/SEA_SURFACE_HEIGHT_ALT_GRIDS_L4_2SATS_5DAY_6THDEG_V_JPL1812 | 4 |
| Monthly Wind Speeds | 0.25 | Monthly | ESA Sentinal-3 Copernicus: https://resources.marine.copernicus.eu/?option=com_csw&view=details&product_id=WIND_GLO_PHY_CLIMATE_L4_REP_012_003 | N/A |

| | | | | |
|---|---|---|---|---|
| Net Primary Productivity (NPP) | 0.25 | Monthly | Vertically-Generalized Production Model (VGPM): http://www.science.oregonstate.edu/ocean.productivity/ | N/A |
| Sea Surface Nitrate (SSN) | 1 | Monthly | World Ocean Atlas 2018 (WO18): https://www.ncei.noaa.gov/access/world-ocean-atlas-2018/ | N/A |
| Mixed Layer Depth (MLD) | 0.5 | Monthly | MIMOC Climatology: https://www.pmel.noaa.gov/mimoc/ | N/A |
| Sea Surface Salinity (SSS) | 0.5 | Monthly | MIMOC Climatology: https://www.pmel.noaa.gov/mimoc/ | N/A |
| Bathymetry | 0.033 | N/A | ETOPO2: https://rda.ucar.edu/datasets/ds759.3/ | N/A |

135

## 2.2 Machine-learning models

We compared the performance of random forest regression (RFR) and artificial neural network (ANN) models at the regional scale. The RFR algorithm is built upon decision tree models, which operate by iteratively generating decision rule nodes that dictate which branch the tree will progress through in the next iteration. The RFR model builds an ensemble, or "forest", of these trees, where each tree is trained on a bootstrapped (*i.e.* randomly subsampled) set of predictors, and the resulting predictions are averaged among the trees to reduce overfitting to noise (Brieman, 2001). In contrast, the ANN model is built as a fully connected network of nodes, or "neurons", in which each neuron consists of an activation function and is connected to other neurons by iteratively-determined weights (Gardner and Dorling, 1998). Both algorithms are advantageous because they make no prior assumptions on the data distributions and can fit non-linear data (Brieman, 2001; Gardner and Dorling, 1998).

Both our ANN and RFR models followed a similar design to Weber et al. (2019). Our ANNs were built using a feed-forward framework consisting of a single input node, two hidden layers each consisting of 30 neurons (using a sigmoidal activation function), and a single output layer (using a linear activation function). A Bayesian L2 (Ridge) regularization parameter was tuned to minimize overfitting and the L-BFGS algorithm was used to solve for weights (Byrd et al., 1995). Each individual decision tree within the RFR was trained using the standard CART algorithm (Brieman, 2001) and constrained to a max depth of 25 decision splits, the simplest configuration determined to perform well and minimize overfitting. These models were built using the Scikit-Learn (v0.24.2) implementation of the ANN ("MLPRegressor") and RFR ("RandomForestRegressor") algorithms in Python 3.8 (see Code Availability).

In both cases, the models were built as an ensemble of either 1000 individual decision trees or individual networks to minimize bias in predictions. The input data were randomly divided for use in model training (80%) and external testing (20%). Although RFR is not sensitive to large differences in predictor variance, predictor data were standardized in both models by normalization to their respective mean and standard deviation. Additionally, we applied an inverse hyperbolic sine (IHS) transformation to the DMS data prior to training (Weber et al., 2019). Testing results indicated that IHS yielded slightly better performance than the more traditional logarithmic transformations for both of our models.

## 2.3 Sea-to-air fluxes

Sea-air DMS fluxes ($F_{DMS}$, µmol m$^{-2}$ d$^{-1}$) were calculated from the monthly-averaged observed and modelled DMS values for June, July and August. $F_{DMS}$ was calculated using the gas transfer velocity ($k$, cm hr$^{-1}$) following the modified approach of Webb et al. (2019):

$$F_{DMS} = k(DMS)(0.24) \tag{1}$$

where the factor of 0.24 converts to the values to daily fluxes. The gas transfer velocity has typically been calculated using a non-linear parameterization (Nightingale et al., 2000), but recent work has suggested a linear parameterization is more appropriate for DMS (Bell et al., 2013; Blomquist et al., 2017; Zavarsky et al., 2018). Since satellite-derived predictors are used to build our models, we calculated the gas transfer velocity using the linear Goddijn-Murphy et al. (2012) k parameterization, which is both derived from satellite altimeter data and normalized to a Schmidt number of 660:

$$k_{w,660} = 2.1U_{10} - 2.8 \tag{2}$$

Where $U_{10}$ is the wind speed (m s$^{-1}$) at 10 m above sea surface.

Regional summertime fluxes ($\bar{F}_{DMS}$, Tg) were calculated as the average ($\pm$SD) quantity of DMS-sulfur emitted over 92 days (June, July and August) through the area of the mapped study region ($1.28 \times 10^7$ km$^2$ or 85.0% of the total bounded area).

## 2.4 Comparison against existing algorithms

Simple linear regression (LR) and multiple linear regression (MLR) models were built for comparison against the machine-learning algorithms. We also tested the performance of our RFR and ANN models against the published algorithms of Simó & Dachs (2002), Watanabe et al. (2007), Vallina & Simó, (2007), and Galí et al. (2018) (hereafter referred to as SD02, W07, VS07, and G18, respectively). Solar radiation dose, SRD, used in the VS07 algorithm was calculated using MLD as described by Vallina & Simó (2007):

$$SRD = \frac{PAR}{K_d \times MLD} \times (1 - e^{-K_d \times MLD}) \tag{4}$$

Each of the four algorithms was assessed using both their original coefficients and coefficients tuned to our NESAP dataset using nonlinear least-squares optimization at both 0.25° and 1° spatial resolution (Table 2). In each case, the algorithms were run using the same monthly-averaged predictors used to develop the RFR and ANN ensembles (see Sec. 2.1). Predictors were spatially matched to either the full DMS dataset (*i.e.* all monthly averaged DMS observations) or to only the Testing partitioned dataset (see Sec. 2.2) for direct comparison with the RFR and ANN ensemble performance (Fig. 2, Table 2).

## 2.5 Controls on DMS variability

Principal component analysis (PCA) was applied to assess the relationships between DMS and the nine predictors used to build the RFR and ANN ensembles. Additionally, non-parametric Spearman rank correlations were calculated between each variable and both the modelled and observed DMS concentrations. Correlation analysis was also extended to assess the role of taxonomy on predicted DMS concentrations, using the outputs of a chlorophyll-a based taxonomic algorithm by Hirata et al. (2011) with NESAP-tuned coefficients (Zeng et al., 2018).

## 2.6 Sensitivity Tests and Predictor Selection

To inform our selection of grid size, we assessed the performance of both the RFR and ANN models using grid cells ranging from 0.25 to 5° (Fig. 1). From this analysis, we found that model accuracy was highest at 0.25° resolution (see Sect. 3.1). Smaller grid sizes would presumably further improve model accuracy, but at a significantly higher computational cost.

We also tested the influence of other biological predictor variables on the performance of the RFR and ANN models, using either NCP, NPP, Chl-a, or PIC. These sensitivity tests indicated no significant difference between the various biological predictor variables, although accuracy was slightly reduced when PIC was used. We therefore selected NCP as the biological predictor variable within our model framework. We also removed $K_d$ as a predictor variable after further sensitivity testing indicated that its exclusion slightly improved results.

The inclusion of nFLH:Chl-a represents a proxy for iron limitation (see Sect. 4.1). However, fluorescence yields corrected for non-photochemical quenching (NPQ) have been suggested to yield a better iron limitation proxy than nFLH:Chl-a (Behrenfeld et al., 2009). We therefore calculated NPQ-corrected fluorescence yields ($\varphi_f$) by:

$$\varphi_f = \frac{nFLH}{Chl-a \times \alpha \times S} \times \frac{iPAR}{\overline{iPAR}} \tag{5}$$

where $\alpha = 0.0147 \times Chl - a^{-0.316}$ and S = 100 mW cm$^{-2}$ µm$^{-1}$ sr$^1$ m as described by Behrenfeld et al. (2009). Our tests indicated nFLH:Chl-a yielded slightly improved performance overall, whereas $\varphi_f$ decreased both models' performance. We therefore retained nFLH:Chl-a and excluded $\varphi_f$ in our final model design.

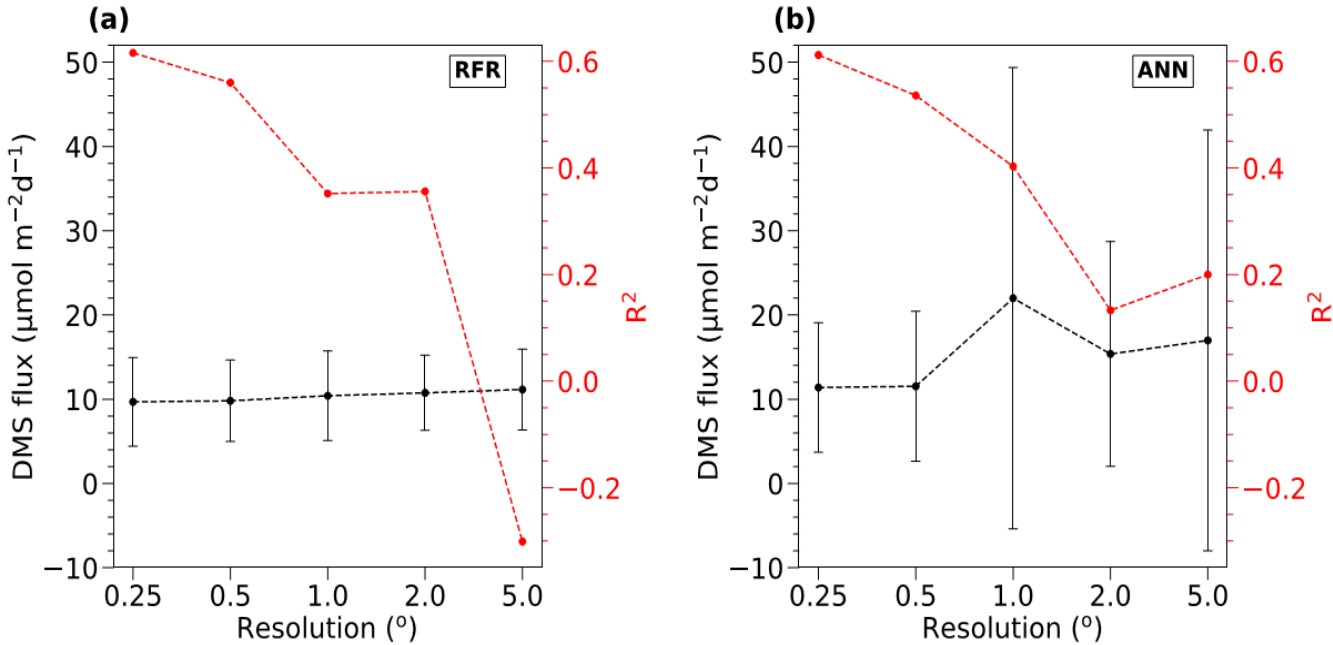


**Fig. 1. Sensitivity of RFR and ANN models to grid size resolution. DMS fluxes (black) and $R^2$ values (red) derived from sensitivity**
**tests of (a) RFR and (b) ANN models to pixels resolutions of 0.25-5º. The negative $R^2$ values observed at the lowest resolution (largest**
**grid cells) indicate that the predicted values explain less variance than the overall mean of the dataset.**

## 3 Results

### 3.1 Model evaluation

To benchmark the performance of our RFR and ANN models, we first evaluated the predictive skill of four
existing empirical DMS algorithms (SD02, W07, VS07, & G18), in addition to simple and multiple linear
regression models. Previous studies have demonstrated that these empirical algorithms show strong predictive skill
($R^2$=0.53-0.84) over large scales and in some oceanic regions (Simó and Dachs, 2002; Galí et al., 2018; Watanabe
et al., 2007), but significantly poorer performance in the NESAP (Herr et al., 2019). Consistent with these results,
we found that the SD02, W07, VS07, and G18 did not accurately predict NESAP DMS distributions, even with
regionally tuned coefficients improving performance (Fig. 2, $R^2$=0-0.01 at 0.25x0.25º; Table 2, r=-0.15-0.36). We
also found that simple and multiple linear regressions performed poorly, yielding virtually no explanatory power
for surface water DMS distributions in the NESAP ($R^2$=0-0.05; Fig. 2, 3).

**Table 2. Performance of statistical DMS algorithms on NESAP DMS observations binned to monthly 1º and**
**0.25º resolution. Pearson correlation coefficients (r) and root mean square error (nM) are obtained from the**

**SD02, VS07, W07 and G18 algorithms (see 2.4) using either their original published coefficients or coefficients derived from non-linear least squares optimization. Algorithm performance is evaluated using either the full monthly-binned observational dataset or using the Testing partitioned dataset (see Sec. 2.2).**

| | SD02 | | VS07 | | W07 | | G18 | |
|---|---|---|---|---|---|---|---|---|
| | Original | Optimized | Original | Optimized | Original | Optimized | Original | Optimized |
| 1°<br>All data | r = -0.09<br>RMSE = 18.03 | r = 0.17<br>RMSE = 4.82 | r = -0.03<br>RMSE = 6.67 | r = 0.03<br>RMSE = 4.96 | r = -0.10<br>RMSE = 11.74 | r = 0.07<br>RMSE = 4.83 | r = 0.02<br>RMSE = 6.77 | r = 0.16<br>RMSE = 4.84 |
| 1°<br>Testing<br>dataset | r = -0.22<br>RMSE = 19.09 | r = 0.36<br>RMSE = 3.34 | r = 0.11<br>RMSE = 5.36 | r = 0.20<br>RMSE = 3.47 | r = -0.03<br>RMSE = 10.46 | r = 0.02<br>RMSE = 3.47 | r = -0.15<br>RMSE = 6.19 | r = 0.30<br>RMSE = 3.40 |
| 0.25°<br>All data | r = -0.05<br>RMSE = 11.02 | r = 0.12<br>RMSE = 7.84 | r = -0.09<br>RMSE = 9.57 | r = 0.11<br>RMSE = 7.88 | r = -0.09<br>RMSE = 13.02 | r = 0.04<br>RMSE = 7.80 | r = 0.06<br>RMSE = 8.42 | r = 0.09<br>RMSE = 7.88 |
| 0.25°<br>Testing<br>dataset | r = -0.03<br>RMSE = 9.79 | r = 0.07<br>RMSE = 6.79 | r = -0.09<br>RMSE = 8.60 | r = 0.10<br>RMSE = 6.79 | r = -0.06<br>RMSE = 12.02 | r = 0.04<br>RMSE = 6.78 | r = 0.04<br>RMSE = 7.47 | r = 0.08<br>RMSE = 6.80 |

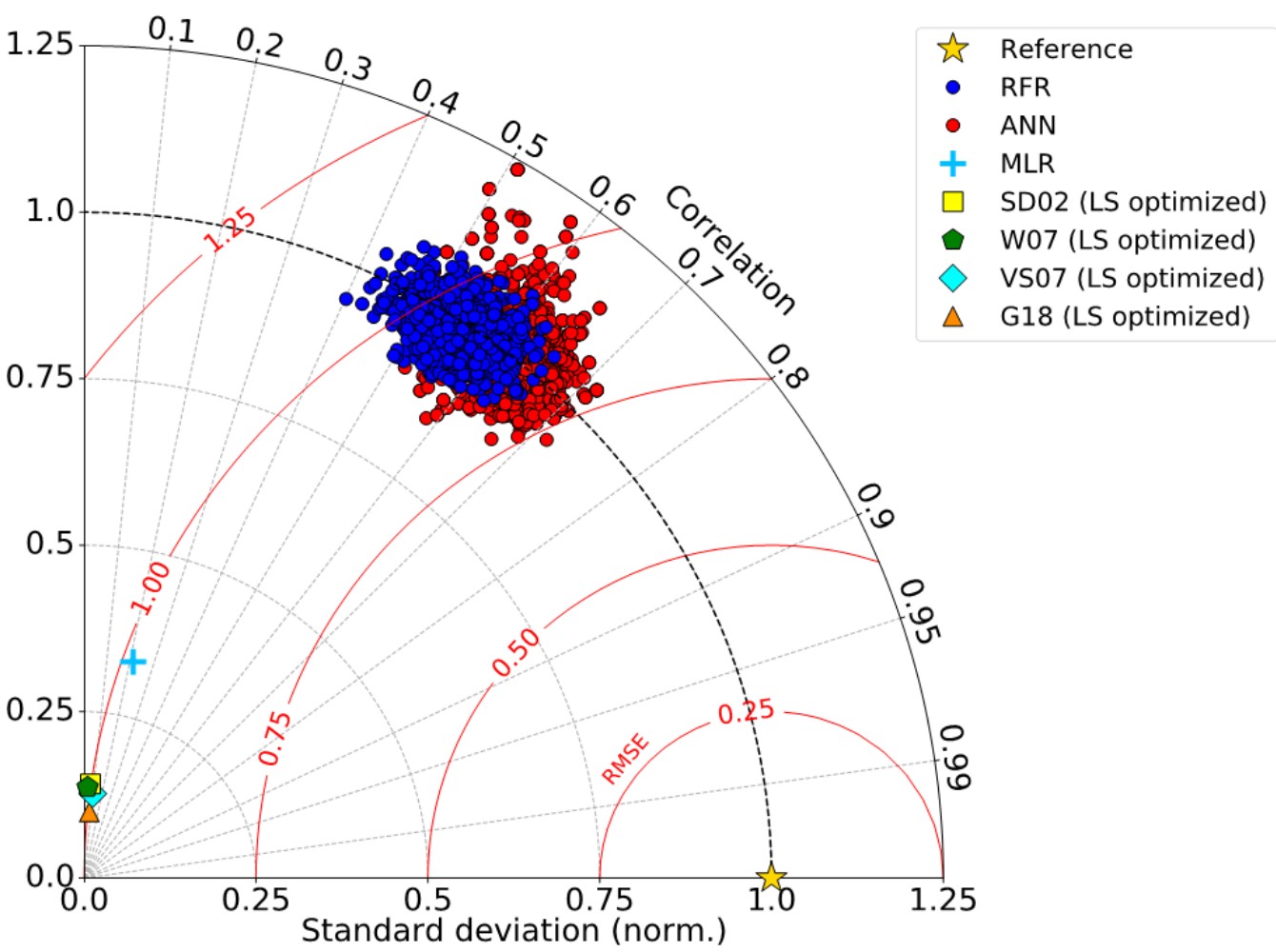

241

**Fig. 2. Taylor Diagram showing comparative performance metrics of each individual Random Forest Regression (RFR)**
**and Artificial Neural Network (ANN) model (1000-model ensembles) against multiple linear regression (MLR) and**
**other statistical DMS models (See sections 2.1 and 2.4). The Pearson correlation coefficients ("Correlation"; outer**
**radius), root mean squared error ("RMSE"; red radial contours), and standard deviations (SDs; grey radial contours**
**from origin) are all computed with respect to the observed DMS samples after inverse hyperbolic sine (IHS)**
**transformation. The reference of a perfect model fit is shown with a gold star. SDs of the model outputs are normalized**
**to the SDs of the DMS observations. RMSE represents a normalized trigonometric derivation from both the correlation**
**coefficients and normalized SDs. Performance of the SDO2, W07, VS07, and G18 algorithms reported here are**
**calculated using regionally tuned coefficients to the NESAP derived from non-linear least-squares optimization (see**
**section 2.4).**

Relative to other published modelling approaches, both the RFR and ANN models dramatically improved
the representation of NESAP DMS variability, achieving significantly higher predictive accuracy (Fig. 2, 3). The
collective ensembles of both the RFR and ANN models yielded strong performance, explaining up to 62% of the
observed DMS variability ($R^2$=0.61-0.62; Fig. 3). For individual models within the ensembles, the ANN method

provided slightly better results ($R^2$=0.16-0.50), compared to the individual RFR models ($R^2$=0.16-0.43). However,
predicted DMS concentrations and sea-air fluxes derived from the ANN ensembles were more sensitive to the
spatial resolution used, although the predictive accuracy of both models degraded significantly with coarser
resolutions (Fig. 1).

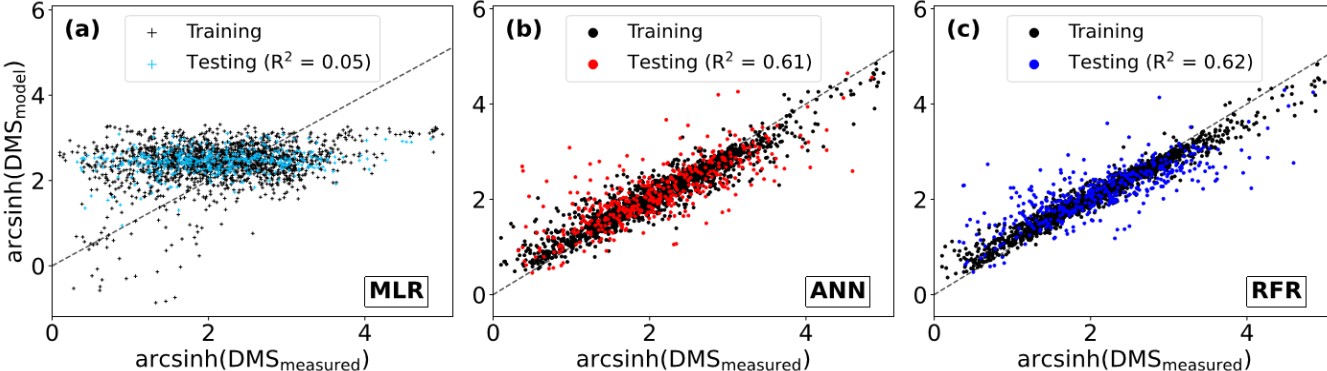


**Fig. 3. Performance of three modelling approaches in predicting observed DMS distributions; (A) multiple linear regression (MLR) (B) ensemble of Artificial Neural Networks (ANN) and (C) ensemble of Random Forest Regression (RFR). For consistency, all predictions are partitioned by the Training and Testing datasets used to build the ensembles (see section 2.2). Model performance ($R^2$) is computed only for the Testing dataset predictions. The dashed line demonstrates a 1:1 relationship. Modelled DMS concentrations depicted range from 0.4-84.3 (RFR, nM) and 0.3-74.6 (ANN, nM).**

## 3.2 DMS distributions and sea-air fluxes

In both the RFR and ANN methods, the predicted spatial distribution of DMS was generally consistent with
observations (Fig. 4a,c,d). The average model derived DMS concentrations was 4.0 ± 2.1 nM and 4.7 ± 3.0 nM
(mean ± SD) for the RFR and ANN ensemble models, respectively, with a similar range from 0.3 to 84.3 nM.  In
both models, the highest DMS concentrations were largely constrained to coastlines and within the Alaska Gyre
adjacent to the Aleutian Islands (Fig. 4b-c, 8C). The greatest discrepancy between DMS concentrations from the
two models was observed in these regional 'hotspots', where the ANN models emphasize high DMS within the
Alaska gyre, while the RFR models emphasize elevated coastal DMS concentrations (Fig. 4b). On average, the
models deviated from each other by 0.49 nM, with the greatest offsets observed in an area of particularly sparse
DMS observations in the Alaska Gyre (Fig. 4a,b). Future observational data in this region should help improve
model agreement.

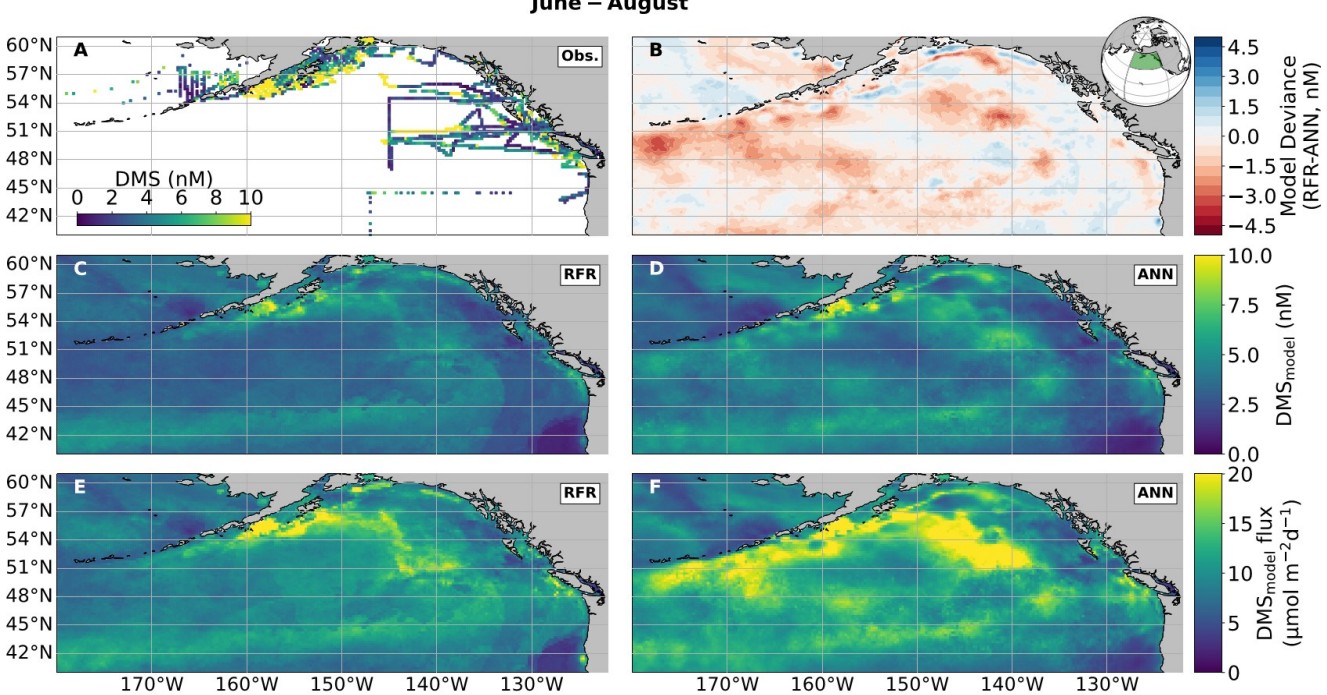

**Fig. 4. Predicted maps of sea surface DMS concentrations and sea-air fluxes. (a) Ship-based observations of mean summertime (June-August) DMS concentrations used to construct the predictive models. (b) Differences between the (c) Random Forest Regression (RFR) and (d) Artificial Neural Network (ANN) ensemble predicted DMS concentrations. (e,f) DMS sea-air fluxes derived from the predicted DMS concentrations. Colormap ranges are restricted to illustrate trends, with <1% of DMS data exceeding the colorbar limits. The inset map in (b) shows the NESAP study region as a shaded green patch in a global orthographic projection.**

Sea-air DMS fluxes (Fig. 4e,f) derived from ANN predictions were 18% higher, on average, than RFR predictions, largely due to higher predicted values in the Alaska Gyre (Fig. 4d-e, Table 3). The distribution of ANN sea-air fluxes was also closer to ship-based observations (Fig. 5). Predicted regional fluxes ranged from 0.8 to 167 $\mu$mol m$^{-2}$ d$^{-1}$ between the two models (Fig. 4e,f, 5), with the highest predicted DMS emissions in August, when derived sea-air fluxes were approximately 1.6 to 2-fold greater than in June and July (Table 3). Our models yielded a summertime integrated sea-air flux of 1.16±1.22 Tg DMS-derived sulfur, which is consistent with the Lana et al. (2011) climatological estimate of 1.64 ± 0.51 Tg (Table 3).


Table 3. Monthly and mean summertime NESAP sea-air DMS fluxes. Total cumulative fluxes of DMS-derived sulfur (Tg,
mean ± SD) are calculated from the Random Forest Regression (RFR) and Artificial Neural Network (ANN) model predictions
(based on an ensemble of 2000 models). Total cumulative NESAP sea-air flux derived from the Lana et al. (2011) climatology
is shown for comparative purposes.

| | RFR | ANN | Summertime Sulfur Emissions | |
| | | | This Study | Lana et al. (2011) |
| --- | --- | --- | --- | --- |
| | µmol m$^{-2}$ d$^{-1}$ | µmol m$^{-2}$ d$^{-1}$ | Tg S | Tg S |
| June | 8.0 ± 5.3 | 8.0 ± 5.5 | 0.29 ± 0.19 | 0.59 ± 0.24 |
| July | 8.2 ± 3.5 | 9.7 ± 4.6 | 0.33 ± 0.14 | 0.41 ± 0.16 |
| August | 12.7 ± 3.5 | 16.5 ± 4.6 | 0.54 ± 0.25 | 0.65 ± 0.25 |
| June-August | 9.7 ± 2.8 | 11.4 ± 4.0 | 1.16 ± 0.35 | 1.64 ± 0.51 |




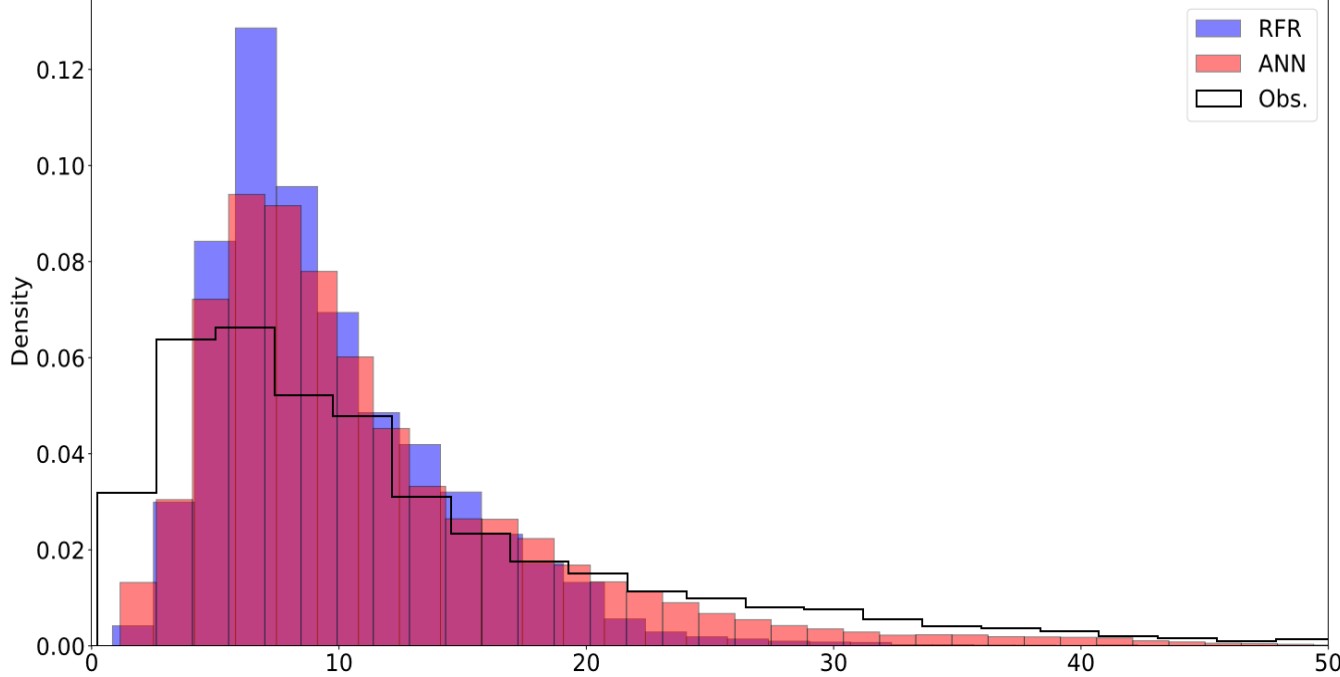


**Fig. 5. Histograms of DMS sea-air flux distributions derived from the 1000-model ensemble random forest regression (RFR) and**
**artificial neural network (ANN) predictions as well as cruise observations (Obs.). The sample sizes of both models are equivalent**
**(n= 49,632) and are significantly higher than the observational dataset (n=2063). Note that the distribution is restricted to show**
**trends, with a maximum flux of 238 nM (Obs.). The upper tail (>50 nM) consists of only 2.9% (Obs.) and <0.1% (both RFR and**
**ANN) of the values. Note that the ANN better predicts the upper tail of DMS observations greater than 20 nM.**

## 3.4 Drivers of DMS variability

In addition to modelling the spatial and temporal distribution of surface water DMS in the NESAP, we
examined the influence of different oceanographic variables as model predictors. As expected based on previous
work (Herr et al., 2019), no single predictor was found to exert a dominant control on modelled DMS distributions
from either the RFR or ANN models (Fig. 6, 7). Rather, the relationship between DMS and other oceanographic
variables exhibited significant region-specific patterns. One of the most compelling regional signatures was the
apparent relationship between DMS and SSHA. In both models, we found significant positive correlations between
DMS and SSHA ($\rho$=0.35, 0.42 for RFR and ANN, respectively) across the full spatial domain, with a particularly
notable relationship along the northern Alaskan coastline (Fig. 8, 9). Here, strong winds (Fig. 9j-l), coupled with
the northeastern Alaska current flow, produce two characteristic oceanographic features in the NESAP: strong,
semi-permanent mesoscale eddies collectively referred to as the Haida, Sitka and Yakutat eddies (Fig. 8a), and the
formation of the high nutrient, low chlorophyll (HNLC) Alaska Gyre (Fig. 8c; Okkonen et al., 2001; Whitney et
al., 2005). Both the monthly (Fig. 9a-i) and summertime-averaged (Fig. 8a,b) RFR and ANN-derived DMS
concentrations are low where these downwelling eddies form. In contrast, elevated DMS concentrations were
associated with the negative SSHA coastal upwelling areas (Fig. 8a,b), where phytoplankton productivity is
stimulated by nutrient inputs into the mixed layer.

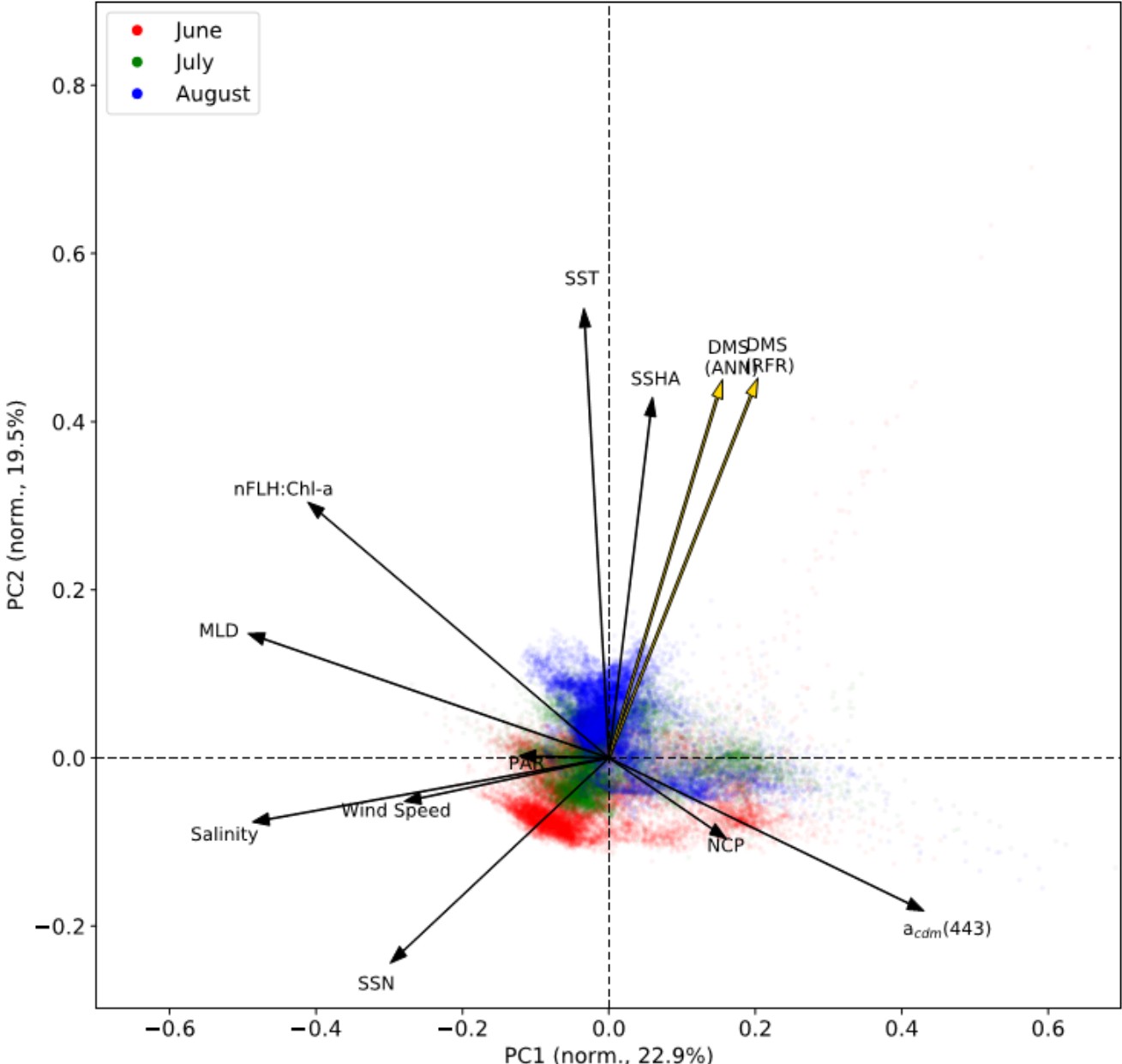


**Fig. 6. Principal Component Analysis (PCA) showing the relationships between variables used to construct the predictive algorithms. Eigenvectors (arrows) are superimposed over the principal components (PCs; data points) for the first two significant modes obtained from PCA. PCs are normalized and clustered by month (June-August, see legend for colors), while the eigenvectors are grouped by ensemble model predictions (gold) and nine predictor variables (black). The percentage of variance explained by each mode is indicated along the axes.**


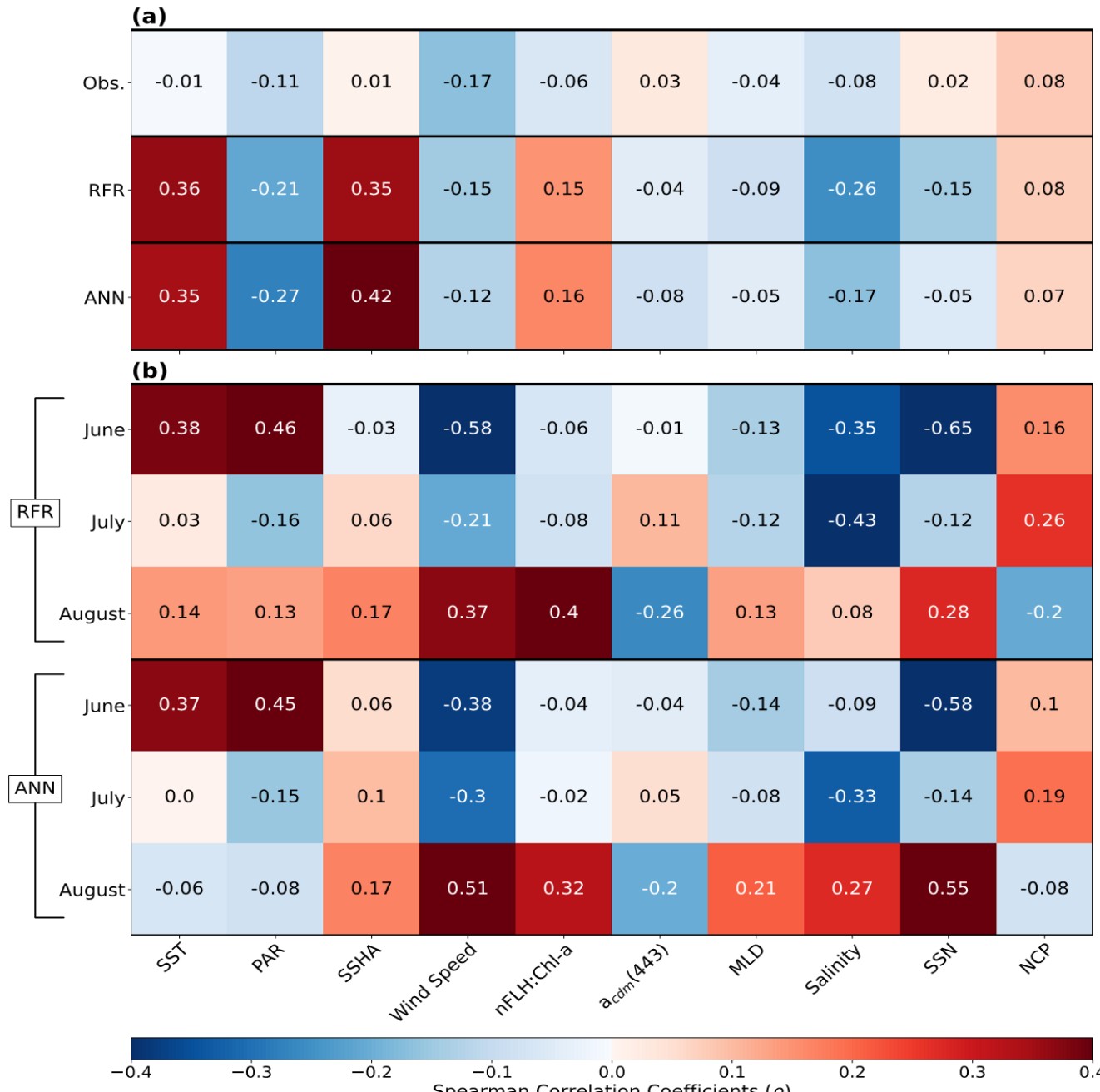


Fig. 7. Heatmap of Spearman rank correlations (ρ). (a) Correlations of pooled data (June-August) for DMS
observations (Obs.), RFR and ANN predictions per variable. (b) Correlations per month for the RFR and ANN DMS
predictions. All model correlations are computed on the 1000-model ensembles.

Modelled DMS concentrations also significantly correlated with hydrographic frontal patterns. We found
significant correlations between DMS and SST ($\rho$=0.36, 0.35 for RFR and ANN, respectively) which suggested
the central Alaska Gyre and offshore of Vancouver Island are areas of elevated DMS variability (Fig. 8b). Both
models predict high DMS levels in the northern frontal zone of the gyre (140°W-145°W) between the 10.5 and
12°C isotherms and the southern frontal zone between (42°N-45°N) between the 13.5 and 15°C isotherms (Fig.
8b,c). By comparison, our models suggest that DMS concentrations are predominantly low in relation to high sea
surface nitrate (SSN) concentrations within the HNLC gyre (Fig. 8, 9). As discussed below, the relationship
between DMS and macronutrient concentrations in the HNLC waters of the central Gulf of Alaska could indicate
an important role for iron limitation as a controlling factor in the DMS cycle. The presence of elevated summer
nutrients in offshore waters is taken as a proxy for iron limitation, which increases over the course of the summer
growing season.

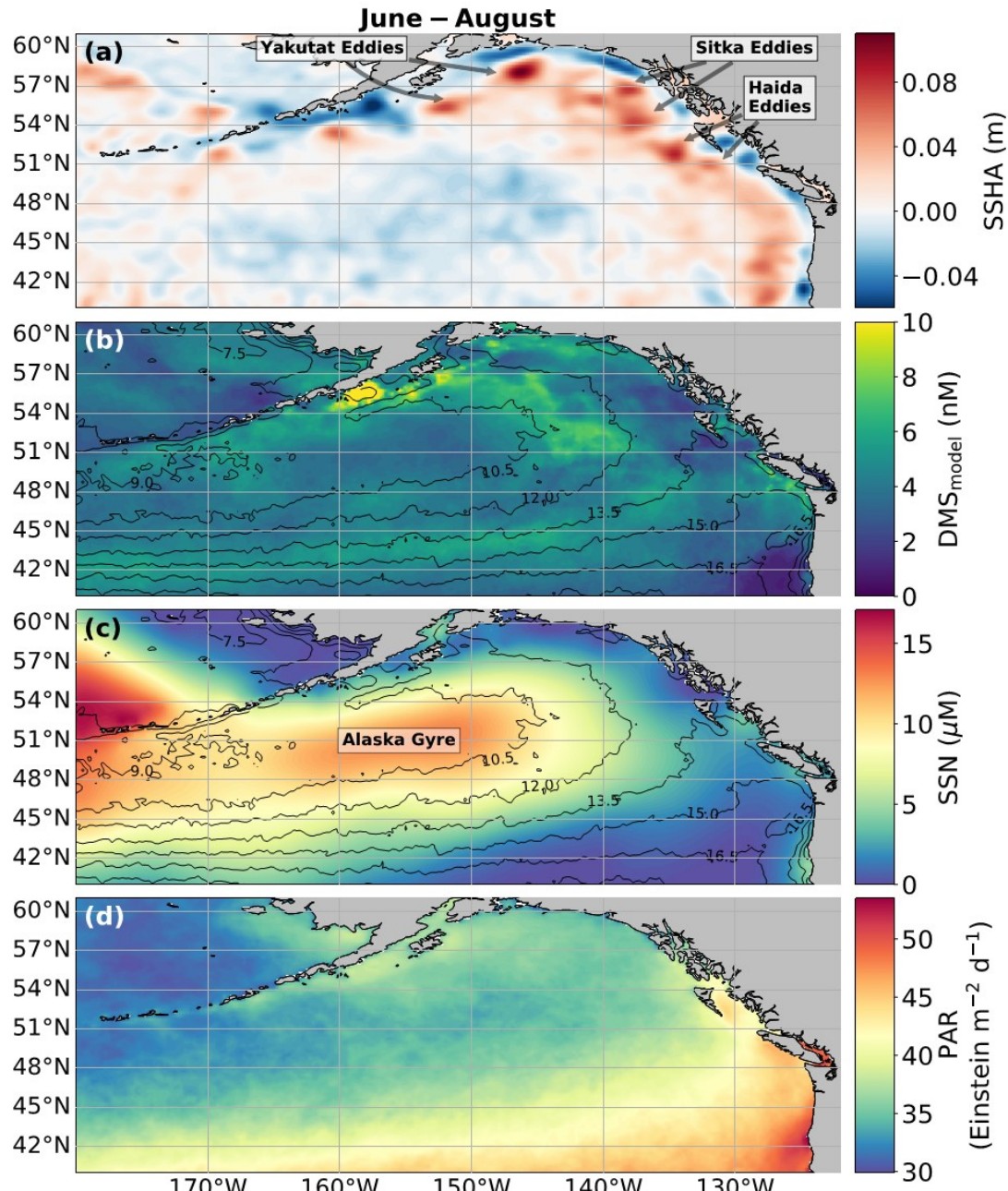


**Fig. 8. Physical drivers of summertime (June-August) NESAP DMS distributions. (a) Sea surface height anomalies (SSHA), (b) predicted DMS concentrations derived from the mean of all 2000 RFR and ANN machine learning models, (c) sea surface nitrate (SSN) and (d) photosynthetically active radiation (PAR). Contours in (b,c) show sea surface temperature (SST) isotherms. Coherent features of elevated sea-surface height indicate the presence of mesoscale eddies, whereas nearshore low SSHAs features reveal areas of upwelling. Colormaps ranges are restricted to illustrate trends with <1% of data exceeding the colorbar limits.**

Other variables appear to exhibit a more localized or minimal influence on DMS cycling. For instance,
both NCP and DMS are elevated in productive nearshore waters, but NCP generally correlates weakly with both
RFR- and ANN-derived DMS concentrations ($\rho$=0.08, 0.07 for RFR and ANN, respectively). It should be noted,
however, the empirically-derived NCP estimates may carry more uncertainty than other predictors obtained from
direct satellite observations (Li and Cassar, 2016). Similarly to NCP, modelled phytoplankton taxonomic
composition (Hirata et al., 2011; Zeng et al., 2018) was not significantly correlated with predicted DMS
concentrations ($\rho$<0.1). Although strong, persistent winds appear to sustain low DMS concentrations off the coast
of Oregon and Vancouver Island (Fig. 9), wind speeds only weakly correlate with DMS overall for the region ($\rho$=-
0.15 and -0.12 for RFR and ANN, respectively). Additionally, high PAR in these areas correspond with low DMS
concentrations (Fig. 6d) and there is an overall negative correlation between PAR and DMS for the region (Fig. 6,
7; $\rho$=-0.21 and -0.27 for RFR and ANN, respectively). Finally, despite hypothesized links between DMS cycling
and iron limitation in the NESAP (Levasseur et al., 2006; Merzouk et al., 2006; Royer et al., 2010), nFLH:Chl-a
ratios (taken as a proxy for phytoplankton iron stress; Behrenfeld et al., 2009; Westberry et al., 2013) did not exhibit
any coherent spatial patterns, and only weakly correlated to our modelled DMS concentrations ($\rho$=0.15 and $\rho$=0.16
for RFR and ANN, respectively).

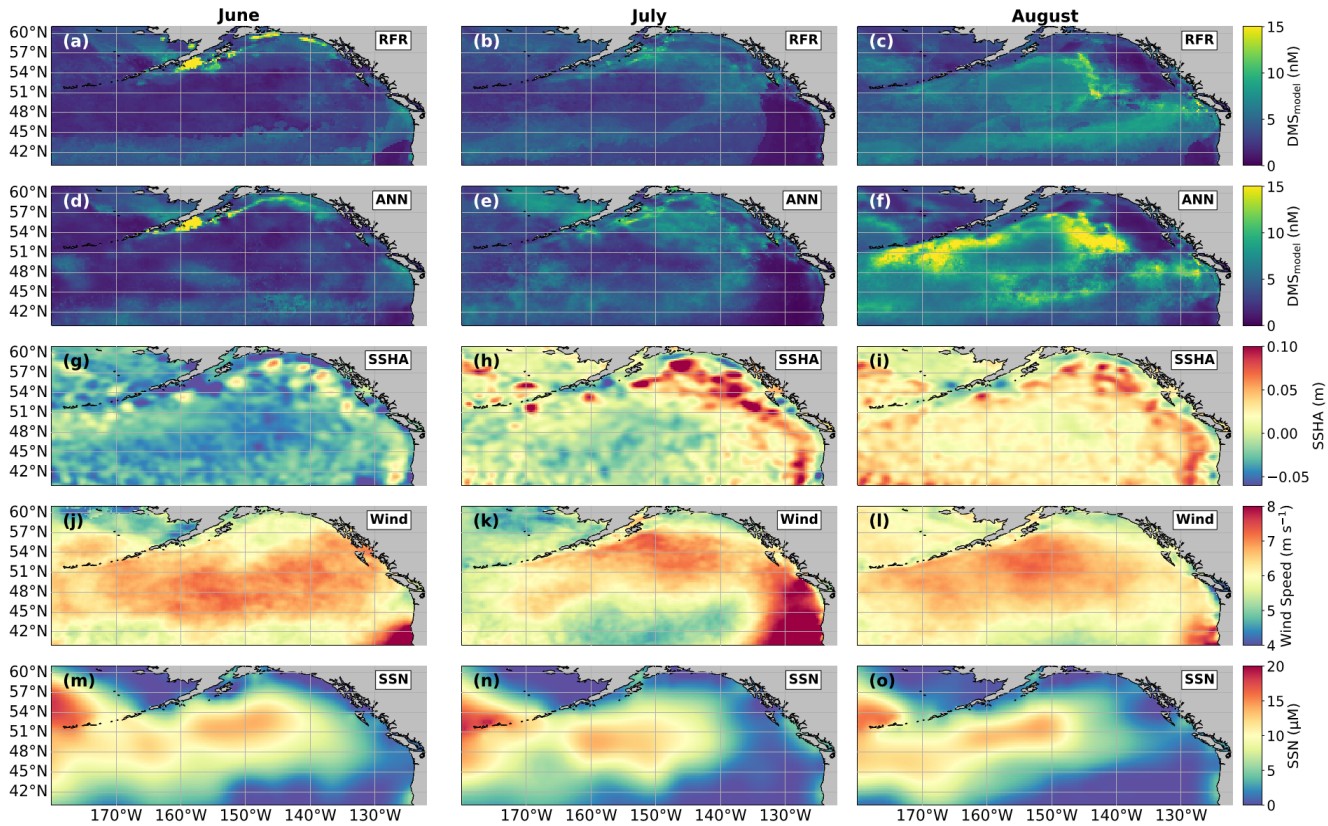

**Fig. 9. Predicted spatial and temporal (June-August) DMS distribution in relation to underlying oceanographic variables. DMS concentrations predicted from (a-c) the Random Forest Regression (RFR) and (d-f) the Artificial Neural Network (ANN) ensemble models are mapped alongside the monthly-averaged (g-i) sea surface height anomalies (SSHA), (j-l) wind speeds (Wind), and (m-o) sea surface nitrate (SSN) for each month. Colormap ranges are restricted to illustrate trends, with at most 1.5% of the data beyond the colorbar limits.**

# 4 Discussion

The relative sparsity of DMS data in many oceanic regions and the complexity of DMS cycling have limited previous attempts to model oceanic distributions of this compound (Simó and Dachs, 2002; Vallina and Simó, 2007; Galí et al., 2018; Watanabe et al., 2007; Herr et al., 2019). Taking advantage of expanding data resources, we employed a new approach to statistically describe DMS distributions in the NESAP. Our results show that both our RFR and ANN models substantially improved predictive strength over traditional empirical approaches (Fig. 2, 3), while identifying several key DMS relationships and regional patterns across the NESAP (Fig. 8, 9). Although our statistical approach does not directly elucidate the underlying mechanisms driving these relationships, and not all variability in predictors may be captured at the single spatial scale used here, we can nonetheless make some

reasonable inductive inferences. These inferences are discussed below, along with the implications of the improved
predictive performance observed here.

## 4.1 Relationships with other oceanographic variables

Among the more prominent spatial relationships we observed was the coherence between predicted DMS
concentrations and SST, and the negative correlation between predicted DMS concentrations and sea surface nitrate
(SSN) within and surrounding the Alaska Gyre (Fig. 6-9). Notably, regional SSN, NCP and Chl-a distributions did
not vary appreciably inside versus outside the gyre, and these variables were poorly correlated with DMS
concentrations (r=-0.02, $\rho$=0.08 with NCP, r=0.09, $\rho$=-0.12 with Chl-a). This suggests that the patterns in surface
DMS across the Alaska Gyre were not simply driven by changes in phytoplankton biomass or productivity. The
DMS-nitrate relationship may be partially explained by the so-called sulfur overflow hypothesis (Stefels, 2000),
which suggests that nutrient-limited phytoplankton increase DMSP production and its subsequent cleavage to
DMS, in order to regulate intracellular sulfur quotas when protein synthesis is limited (Hatton & Wilson, 2007;
Kinsey et al., 2016; Simó & Vila-Costa, 2006; Spiese & Tatarkov, 2014; Stefels, 2000). This mechanism may help
explain the higher predicted DMS concentrations at the northern extent of the Alaska Gyre, where SSN
concentrations begin to decrease (Fig. 6). Nutrient-dependent effects may also be important in explaining seasonal
variability, as the DMS-nitrate relationship becomes positive in August as phytoplankton growth becomes
increasingly nutrient limited (Fig. 7b).
The apparent relationship between DMS and nitrate could also result indirectly from the underlying effects of
iron limitation. Excess summertime nitrate concentrations are taken as evidence for iron limitation in the NESAP
(Boyd and Harrison, 1999; Boyd et al., 2004; Martin and Fitzwater, 1988; Whitney et al., 2005). Under iron-limiting
conditions, DMS is thought to function, together with DMSP and DMSO, as part of an antioxidant response to
oxidative stress (Sunda et al., 2002). This hypothesis suggests that iron limitation should stimulate net production
of DMS and DMSP (Bucciarelli et al., 2013; Sunda et al., 2002), which is inconsistent with the overall negative
dependence predicted between DMS and SSN (Fig. 8b,c).
Satellite-based, chlorophyll-normalized fluorescence has been suggested as an additional proxy for iron
limitation. Low iron conditions can lead to both a reduction in photosystem I relative to photosystem II (Strzepek
and Harrison, 2004), and an apparent increase in energetically-decoupled light harvesting complexes (Allen et al.,
2008; Behrenfeld & Milligan, 2013), resulting in elevated fluorescence-to-chlorophyll a ratios (nFLH:Chl-a)
(Westberry et al., 2013). To our knowledge, this proxy has not been widely investigated with respect to DMS
cycling. In our analysis, we found that nFLH:Chl-a ratios, and the NPQ-corrected fluorescence yields ($\varphi_f$), exhibited

only weak positive correlations with the RFR and ANN predicted DMS concentrations (Fig. 6, 7). Moreover, neither of these metrics exhibited coherent spatial patterns with predicted DMS concentrations, suggesting a limited role for iron in driving spatial patterns of DMS cycling within the NESAP. However, it is important to note the potential temporal mismatch between our monthly DMS predictions and these more instantaneous metrics of iron limitation, which reflect short-term physiological changes (days to weeks; Behrenfeld et al., 2009; Westberry et al., 2019) that depend on sporadic iron loading (*e.g.* aerosol deposition; Mahowald et al., 2009). Indeed, both natural and artificial iron-fertilization events have thus far been detected from satellite-derived nFLH:Chl-a at daily resolution (Westberry et al., 2013), in contrast to the monthly-averaged data used here. Therefore, modelling frameworks utilizing shorter temporal scales may find a clearer connection between DMS cycling and iron limitation using the chlorophyll-a fluorescence proxy.

Beyond nutrient limitation effects, ambient light fields are believed to exert significant direct and indirect effects on DMS cycling (del Valle et al., 2007). At the community level, high irradiance may inhibit bacterial consumption of DMS (Slezak et al., 2001; Toole et al., 2006; Lizotte et al., 2012), while covarying changes in mixing and high irradiance can induce transient selectivity for high-light acclimated species and influence the proportion of  high DMS/P producers within assemblages (Galí et al., 2013; Vance et al., 2013). Ultraviolet radiation has been noted to induce high DMS production and turnover through a proposed cascading oxidation pathway, which acts to remove harmful reactive oxygen species (Sunda et al., 2002; Archer et al., 2010). In contrast, more recent evidence has indicated the potential for elevated DMS production in the NESAP from the reduction of DMSO due to light-induced oxidative stress over diurnal cycles (Herr et al., 2020). Although our modelled DMS concentrations exhibited an overall negative correlation with PAR (Fig. 6, 7a), monthly correlations indicate a stronger positive correlation between DMS and PAR in June, where the summer solstice drives high irradiance. In contrast, July and August exhibit much weaker negative correlations as the summer bloom declines (Fig. 7b). These results provide indirect evidence that light-induced oxidative stress, possibly coupled with inhibition of microbial DMS consumption, may influence regional NESAP DMS distributions, particularly early in the summer.

The overall negative association of DMS and incident light (Fig. 6,7a) may also indicate a role for photolysis in DMS loss through (del Valle et al., 2007). Since DMS does not have strong light absorption properties, the presence of photosensitisers is necessary for the abiotic photooxidation of DMS (Brimblecombe and Shooter, 1986). To account for this process, our models incorporated nitrate (SSN) and $a_{cdm}$(443) (as a proxy for CDOM; Nelson & Siegel, 2013), both of which are thought to be dominant photosensitisers of DMS in marine systems (Taalba et al., 2013; Bouillon and Miller, 2004, 2005; Galí et al., 2016). In the NESAP, nitrate appears to exert a stronger influence than CDOM on the apparent quantum yields (AQY) of DMS (Bouillon and Miller, 2004). In

support of this, our results suggest a stronger negative dependence of predicted DMS concentrations on nitrate
compared to CDOM within the NESAP, particularly in June when irradiance is high (Fig. 6, 7). We acknowledge,
however, that the DMS-nitrate relationship likely also reflects physiological impacts of nutrient limitation, as
discussed above. Nonetheless, our results are consistent with elevated rates of DMS photo-oxidation in the nitrate-
replete, low iron waters of the Alaska Gyre, where photolysis may drive strong DMS oxidation and explain the low
predicted DMS concentrations (Fig. 8, 9). Further *in situ* work will be required to resolve the relative contributions
of these biotic and abiotic processes to DMS cycling within these areas.
Among all the statistical relationships we observed, perhaps the most striking was the association of DMS
variability with SSHA, particularly along the Alaskan coast and in relation to mesoscale eddies (Okkonen et al.,
2001; Whitney et al., 2005; Fig. 8, 9). To our knowledge, only one other study has linked SSHA to DMS within
the NESAP. Herr et al., (2019) demonstrated contrasting positive and negative correlations between DMS and
SSHA in offshore and coastal waters, respectively, in general agreement with our results. Presently, the underlying
mechanisms explaining the relationship between SSHA and DMS cycling remain unclear, yet it is likely that
physical mixing processes are important. For example, enhanced biological production is known to be stimulated
by eddy re-supply of iron and macronutrients via vertical advection and diffusion (Whitney et al., 2005; Bailey et
al., 2008). These nutrient supply processes would also be expected to influence DMS cycling, as outlined above.
Elevated abundances of high DMS-producers within anticyclonic eddies with positive sea surface height anomalies
have been noted in the Sargasso Sea (Bailey et al., 2008), while eddy-induced vertical transport likely supplements
nearshore, current-driven upwelling that can also resupply iron into the coastal waters of the NESAP (Cullen et al.,
2009; Freeland et al., 1984). In addition, eddy propagation can allow cross-shelf transport, distributing
micronutrients to offshore waters (Fiechter and Moore, 2012), potentially contributing to the apparent elevated
DMS concentrations in the outer Alaska gyre between the 10.5 and 12°C isotherms (Fig. 8). These mixing and
transport mechanisms could partially explain the influence of elevated productivity in driving increased nearshore
and northern NESAP DMS concentrations (Fig. 4, 7-9), representing a novel source of DMS variability in this
region.
The taxonomic composition of plankton assemblages is also a likely source of variability influencing DMS
cycling. Significant changes to DMS production and consumption rates within the NESAP are expected in response
to variable microbial and phytoplankton taxonomy (Vila-Costa et al., 2006; Lidbury et al., 2016; Sheehan and
Petrou, 2020). Such taxonomic variability may, in turn, reflect transient community composition shifts in response
to mixing (Bailey et al., 2008), nitrate (Bouillon and Miller, 2004), and iron availability (Levasseur et al., 2006;
Merzouk et al., 2006). The monthly averaging used in our data processing removes autocorrelation associated with
individual sampling expeditions (Wang et al., 2020), but it may preclude capturing these transient taxonomic
responses. For instance, coccolithophores are believed to influence DMS cycling in the NESAP (Herr et al., 2019;
Asher et al., 2011), yet monthly-averaged calcite distributions did not yield increased predictive strength for DMS
concentrations in our analysis (see Sect. 2.6). However, as satellite PIC preferentially reflects the optical signature
of detached coccoliths, monthly-averaged satellite PIC observations may represent the senescence of
coccolithophore blooms, rather than active growth phases. Additionally, applying a chlorophyll-a based taxonomic
algorithm (Hirata et al., 2011; Zeng et al., 2018) yielded no further explanation of the DMS variability predicted.
The influence of taxonomic composition thus remains cryptic within our modelling framework.
**4.2 Implications of Improved Predictive Power**
As noted above, both the RFR and ANN approaches demonstrate significantly improved accuracy over
existing models, explaining up to 62% of observed DMS variability (Fig. 2, 3). This predictive skill is somewhat
lower than that achieved for methane fluxes (Weber et al., 2019) and dissolved inorganic carbon dynamics (Roshan
and DeVries, 2017), where $R^2$ values ranging from 0.7 to 0.95 were obtained. Nonetheless, the dramatic accuracy
improvement of our algorithms over traditional methods (Fig. 2, 3) encourages the further use of these techniques
in modelling DMS distributions.
Improved predictive accuracy provides opportunities to gain insight into the mechanisms driving DMS cycling.
Our approach has yielded accurate DMS predictions at a 4 to 40-fold higher resolution then previous algorithms
(Simó and Dachs, 2002; Vallina and Simó, 2007; Galí et al., 2018; Watanabe et al., 2007), enabling the description
of mesoscale patterns and processes (Fig. 8). Extending these methods to sub-mesoscale resolution will enable
investigations into the dependence of DMS on finer-scale hydrographic processes, particularly stratification and
frontal dynamics, which have been increasingly linked to DMS cycling but remain unresolved mechanistically
(Royer et al., 2015; Asher et al., 2011). Moreover, coupling machine learning algorithms with biophysical and
tracer export models holds promise to resolve the contributions of eddy dynamics and upwelling intensity on DMS
variability, likely through nutrient availability and physiological mechanisms (Asher et al., 2011; Bailey et al.,
2008; Cullen et al., 2009). Recent work has also developed a new database of DMS apparent quantum yields (Galí
et al., 2016). As the availability of these measurements increases, simultaneous mapping of both DMS quantum
yields and concentrations will become feasible, enabling future studies to better parse out the contribution of
photolysis, physical mixing, and biological drivers of DMS cycling.
Although used in a diagnostic capacity here, our statistical models also hold potential for prognostic
applications. Frameworks utilizing shorter time scales will likely be able to detect underlying mechanisms
driving observed diel cycling (Galí et al., 2013; Royer et al., 2016), even if the underlying mechanisms are still
unresolved. We note, however, that caution will need be exercised as machine learning models have a tendency
to overfit noise (Weber et al., 2019; Roshan and DeVries, 2017; Wang et al., 2020), thus requiring appropriately
large training datasets and the use of known "future" observations to validate predictive accuracy in this context.
The significant variability in DMS cycling across oceanic regimes will likely also render predictions more
successful at regional, rather than global, scales (Galí et al., 2018; Royer et al., 2015). Nonetheless, prognostic
applications of these algorithms should be investigated to aid in the future development of improved mechanistic
models.

# 5 Conclusions

We have presented a statistical approach for modelling DMS distributions, which provides significantly
higher predictive skill than traditional methods (Simó and Dachs, 2002; Vallina and Simó, 2007; Galí et al., 2018;
Watanabe et al., 2007; Lana et al., 2011), and yields estimates of the summertime NESAP DMS sea-air fluxes to
1.16±1.22 Tg S in agreement with previous findings (Herr et al., 2019; Lana et al., 2011). Our results further
underscore the importance of the NESAP to global DMS production and motivate further observations in
traditionally under-sampled areas such as the Alaska Gyre and Aleutian Islands. Although we are unable to directly
examine the mechanistic drivers of DMS variability, our findings suggest nutrient limitation, light-driven
processes, and eddy-induced mixing are potentially key drivers of DMS cycling in the NESAP. Future studies will
benefit from using such statistical algorithms, in conjunction with field-based process studies and mechanistic
models, to better understand the underlying dynamics and driving factors in the oceanic DMS cycle.
*Code availability.* The analysis in this study makes extensive use of the Numpy, Matplotlib, & Scikit-Learn libraries
in      Python.      The      custom      codes      used      can      be      downloaded      at
https://github.com/bjmcnabb/DMS_Climatology/tree/main/NESAP or are available upon request from the
corresponding author.
*Data Availability.* DMS observations and predictor datasets are described in the Methods with relevant links to
repositories. Data from the Lana et al. (2011) climatology used for comparison in Table 3 are available via the
SOLAs project (retrieved from www.bodc.ac.uk/solas_integration/implementation_products/group1/dms/), where
the DMS sea-air fluxes were calculated as described in Sect. 2.3. The gridded climatologies produced from each
algorithm in this study can be obtained at
https://github.com/bjmcnabb/DMS_Climatology/tree/main/NESAP/Climatologies.
*Author Contribution.* BM and PT designed the study. Model code was written and implemented by BM. BM
prepared the manuscript with significant contributions from PT.
*Competing Interests.* The authors declare that they have no conflict of interest.
*Acknowledgements.* We would like to thank Dr. Valentina Radic for her advice in model design and four
anonymous reviewers for their helpful comments that improved this manuscipt. This work was supported by grants
to BM and PT from the Natural Sciences and Engineering Research Council of Canada (NSERC).

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
