# Peer review of "Improved Prediction of Dimethyl Sulfide (DMS) Distributions in the"

_Biogeosciences, 2021_

## Author Response (AR1)

**Reviewer #1**

I was particularly glad to find that the authors evaluated the optimal spatial scale for the ML predictions. However, and this is my first criticism, they did not perform a similar evaluation for the temporal resolution of the data, and used only monthly-binned DMS datasets. My questions: How compatible is binning to monthly resolution with the claim that your ML methods can resolve DMS concentration at the mesoscale? I.e., is this temporal resolution sufficient? Changes in DMS cycling regimes often match the timescales of meteorological forcing, i.e. days to weeks (Royer et al., 2016), and several meteorological re-analysis products are readily available as inputs for the analysis of the optimal temporal scales of the ML models.

*As discussed in this paper, much of the emergent mesoscale patterns in surface DMS distributions result from persistent oceanographic features, such as eddies and hydrographic frontal zones. These features are well resolved in monthly averaged data. Although some transient temporal variability may be obscured by monthly binning (as mentioned on L444-455), our goal here was to identify persistent summertime DMS features within the NESAP that are comparable to previous monthly climatological work.*

*Additionally, several technical factors influenced our decision to use monthly resolution. Most notably, training these models on daily, 5, or 8 day resolved observations introduces autocorrelation among observations from the same cruise, which can bias the resulting predictions (L426-427, see discussion in Wang et al. 2020). Monthly binning of the available DMS observations allows us to reduce this source of spurious correlation. Additionally, data products for certain predictors (e.g. MLD, SSN) are not currently available at higher temporal resolution. Lastly, monthly averaged data also have less uncertainty associated with interpolation through areas of heavy cloud cover.*

My second general criticism concerns the comparison between the ML models and previous DMS algorithms based on simpler traditional statistics. Such comparison would be useful to readers if conducted differently, but in its present form it is too shallow, and its sole purpose seems to be highlighting the better performance of the ML models (which one can take for granted, as shown by abundant recent literature on the subject). In my view, each approach has its pros and cons, and both of them should be included in a fair evaluation. First, in my view the predictive power of the different approaches should be compared at the same spatiotemporal scale. Second, the regional tuning of the global algorithms shown in Fig. 2 should be described more in depth somewhere, not just referring to Table 5 of Herr et al. (2019). Tuning each of the previous algorithms for a particular region is a complicate task in itself. Table 5 shown in Herr et al. 2019 made evident that the tuning they applied was not always effectively improving the algorithms: it improved some skill metrics (e.g., correlation) at the expense of other (e.g., RMSE degraded notoriously in many cases). Third, it would also be interesting to learn why regionally tuned algorithms that explained nearly 10% (VS07) or 7% of the DMS variance in the NESAP now appear to explain less than 1%. I do not think that the datasets used by Herr et al. (2019) and this paper are that different… is the difference only due to the finer resolution used in the present study?

*We have clarified and expanded the methods (L169-173) and results (L212, Table 2 in L216-221) to address these comments. In short, each of the four algorithms was run using the same monthly resolution at 0.25x0.25º, and we have now added a table (see Table 2) including a comparison of algorithm performance between 0.25x0.25º and the more traditional resolution of 1x1º. Each algorithm was run on using all the observations, and also with only the testing portion of the dataset*

*for direct comparison with the RFR and ANN models. In all cases, model fit (indicated by correlation coefficients and RMSE, see new Table 2) was improved with the application of non-linear least squares optimization. Note that although the DMS dataset used in this study is indeed very similar to that used in Herr et al. (2019), their dataset included observations from 1984-1997 that were omitted here to better match available satellite/climatological predictors. Additionally, we used a different source of MLD and nitrate data than Herr et al.*

My third general criticism concerns the way regionally aggregated emissions are reported (Table 2; line 20 in the abstract; lines 258–263 in the Results). First, it is incorrect to call "annual sulfur emissions" what actually are summertime emissions.

*We have corrected Table 3 to reflect the fact that these values represent only summertime emissions.*

Second, the total regional summertime emissions cannot be the mean of the three individual months (0.3 Tg S) but their addition. Only in this way the range reported in the abstract, 0.5–2 Tg S (per year? Per summer?) can be compatible with the monthly emissions reported in Table 2. Finally, the authors must explain how they obtained the uncertainty range of 0.5–2 Tg S per year (but, do they refer to summer only or the whole year?) given in the abstract and in line 158.

*We have updated the total emissions (Tg) for this study and those derived from Lana et al. (2011) to represent the sum of the fluxes from June to August in Table 3.*

*The uncertainty range (in Tg S $yr^{-1}$) initially listed in the abstract and line 158 (± one standard deviation) was derived from first computing the averaged sea-air fluxes for the region (Tg S from DMS from both the RFR and ANN predictions) and then scaling these values to a summertime-equivalent annual flux using the fraction of days modelled out of the year (365/92). This calculation assumed that the majority of DMS emitted yearly in the NESAP is during June to August, to coincide with the peak of the growing season.*

*Upon reevaluation, we note this approach likely provides an erroneous estimate. As a result, we have now removed these annual flux estimates, and retained only the calculated summertime total, DMS-derived S flux (Tg S). Likewise, the comparison to global uncertainties in yearly rates (L289-291) has been removed.*

Finally, I prompt the authors to improve the Discussion. I provided some ideas in the specific comments in the hope that they will be useful.

*Thank you for your thoughtful comments, we have detailed below the revisions made throughout.*

**Specific comments**

Introduction

L25-26: a more up-to-date reference would be good here.
*We have added the following reference here:*

*Ksionzek, K. B., Lechtenfeld, O. J., McCallister, S. L., Schmitt-Kopplin, P., Geuer, J. K., Geibert, W., and Koch, B. P.: Dissolved organic sulfur in the ocean: Biogeochemistry of a petagram inventory, Science, 354, 456–459, https://doi.org/10.1126/science.aaf7796, 2016.*

L29: DMS does not seem to be an essential substrate for most pelagic prokaryotes, but for rather specialized methylotrophic taxa, as suggested by Vila-Costa et al. 2006. Most taxa do not seem to use it as a carbon source, and the enzymes that degrade it might by be quite unspecific. Please rephrase with additional supporting references. Suggestions:

• Schäfer, Hendrik, Natalia Myronova, and Rich Boden. "Microbial degradation of dimethylsulphide and related C1-sulphur compounds: organisms and pathways controlling fluxes of sulphur in the biosphere." Journal of experimental botany 61.2 (2010): 315-334.

• Green, David H., et al. "Coupling of dimethylsulfide oxidation to biomass production by a marine flavobacterium." Applied and environmental microbiology 77.9 (2011): 3137-3140.

• Hatton, Angela D., et al. "Metabolism of DMSP, DMS and DMSO by the cultivable bacterial community associated with the DMSP-producing dinoflagellate Scrippsiella trochoidea." Biogeochemistry 110.1 (2012): 131-146.

• Lidbury, Ian, et al. "A mechanism for bacterial transformation of dimethylsulfide to dimethylsulfoxide: a missing link in the marine organic sulfur cycle." Environmental microbiology 18.8 (2016): 2754-2766.

*Thank you for the suggestions. We have rephrased this section and included further supporting references to clarify the importance of DMS as an important substrate for particular planktonic groups.*

L75-78: I missed here references to 2 important studies in the NESAP:

• Royer, Sarah-Jeanne, et al. "Microbial dimethylsulfoniopropionate (DMSP) dynamics along a natural iron gradient in the northeast subarctic Pacific." Limnology and oceanography 55.4 (2010): 1614-1626.

• Steiner, Nadja S., et al. "Evaluating DMS measurements and model results in the Northeast subarctic Pacific from 1996–2010." Biogeochemistry 110.1 (2012): 269-285.

These studies suggested an important role for phytoplankton community species composition (prymnesiophytes, dinoflagellates) and higher bacterial DMS yields from dissolved DMSP in Fe-poor offshore NESAP waters.

*Thank you for suggesting these papers, we have rephrased L75-78 to include these.*

Methods

L103: Please specify NASA reprocessing (e.g., R2018).

*We have specified that the R2018 data products were used in this study.*

L103: the resolution looks wrong. I am not aware of any NASA product with 0.036 degrees resolution. The 1/24th degree resolution probably used here corresponds to ~0.042 degrees.

*Thank you for noticing this discrepancy, we have corrected the resolution for Aqua MODIS data throughout. We have also specified the resolutions for SeaWiFS and Aqua TERRA data when used.*

L137-138: A short discussion of this finding might be useful for future studies.

*The choice of IHS transformation was largely influenced by the work of Weber et al (2019) (a citation has been added here). As noted in text, IHS was used because it produced marginally improved accuracy compared to log transformation, but the differences were not significant.*

Figure 1: please use the same left and right y-axes in both panels.

*We have updated these y-axes scales to match.*

Results

L208-209: R2 reported twice, the range given in L208 suffices.

*We have removed the $R^2$ reported in L209.*

L226: I understood from the Methods that the ML models were used to estimate sea-surface DMS concentration, not sea-air fluxes directly. This would make more sense because sea-air fluxes are derived from a known parameterization. Therefore, is the sentence "the models showed lower predictive power for sea-air DMS fluxes at coarser resolution (Fig. 1)" accurate?

*We have rephrased the language here to better represent the methodology used.*

Figure 3: if DMS is arcsinh-transformed, the nM units no longer make sense. Either modify the axes to show actual DMS concentrations, or remove the nM units. In the latter case, it would be useful for the readers to know the range of DMS shown in the scatterplots.

*We have removed the nM units from the axes and provided the range of DMS concentrations predicted from the RFR and ANN models.*

L258-263: see general comment about the regional emissions.

*Please see response comments above.*

L261: How was this range obtained? It seems that the actual range resulting from the combination of regional and global uncertainty ranges should be larger, assuming their errors are uncorrelated. For example, 0.5/28 gives a minimum of ca. 2%, 2/15 gives a maximum of 13%...

*The range was reported as one standard deviation above and below the mean. As mentioned above, we have now removed this comparison, as the annual fluxes initially reported were likely erroneous.*

L303: why the central Alaska gyre, and not other subregions within the NESAP?

*We have rephrased L323 to note that the offshore waters of Vancouver Island also show elevated DMS concentrations associated with SST fronts (Fig. 8).*

L318-322: correlations between DMS and other variables that can be directly observed (SST, SSHA) are not comparable to those between DMS and variables output by empirical or prognostic models, such as NPP (VGPM) or NCP. This should be mentioned, as the skill of that latter models is quite limited in some regions.

*We now mention the uncertainty associated with NCP estimates within this study in L424-426.*

L327: Royer et al. 2010 L&O, too.

*We have added this reference.*

Discussion

L352-355: The sulfur overflow hypothesis may describe sulfur metabolism in species with high intracellular DMSP concentration and where DMSP is the main sulfur osmolyte. This hypothesis may not be relevant for species with low intracellular DMSP that produce other sulfur metabolites in similar or higher quantities, e.g. DHPS (Durham et al., 2015). Might this be the case for the northern part of the Alaska gyre?

•   Durham, Bryndan P., et al. "Cryptic carbon and sulfur cycling between surface ocean plankton." Proceedings of the National Academy of Sciences 112.2 (2015): 453-457.

*We considered DHPS as a contributing factor but feel that there is currently insufficient evidence to suggest that this compound is linked directly to oceanic DMS distributions.  We note that Durham et al. (2019) showed that DHPS was primarily produced by diatom-rich coastal communities, which are also unlikely to dominate assemblages within the gyre where predicted DMS concentrations are low.*

Moreover, total phyto biomass is, per se, a strong predictor of DMS outside the subtropical and intertropical areas, and is strongly negatively correlated to nitrate I guess… this would give a more straightforward explanation of the negative DMS-nitrate correlation found here.

*We investigated biomass as the source of the DMS-nitrate relationship here but found that chlorophyll-a was not strongly correlated with nitrate (r=0.09, ρ=-0.12) nor did it appreciably differ inside vs outside the gyre. We have now added these considerations to this section in L375-378.*

L380-383: The authors mention only physiological aspects of the effects of (UV) irradiance on DMS cycling. Please consider giving a wider view that considers community-level effects, for which ample evidence exists, for example:

•   Lizotte, Martine, et al. "Macroscale patterns of the biological cycling of

dimethylsulfoniopropionate (DMSP) and dimethylsulfide (DMS) in the Northwest Atlantic." Biogeochemistry 110.1 (2012): 183-200.

• Galí, Martí, et al. "Diel patterns of oceanic dimethylsulfide (DMS) cycling: Microbial and physical drivers." Global Biogeochemical Cycles 27.3 (2013): 620-636.

• Vance, Tessa R., et al. "Rapid DMSP production by an Antarctic phytoplankton community exposed to natural surface irradiances in late spring." Aquatic microbial ecology 71.2 (2013): 117-129.

*We have expanded this section (L540-L543) to discuss community level effects.*

L386-387: Correlation analysis between DMS and potential predictor variables should perhaps consider the different spatiotemporal scales of variation each variable is capturing. For example, in the case of PAR, the negative correlation might arise solely from the fact that DMS peaks in August-September (as does Chl), whereas the highest PAR is in June (summer solstice). The kind of information we obtain from the DMS-PAR correlation is very different from that provided by SSHA, which informs mostly about mesoscale variability, or SSN, which reflects circulation patterns. Therefore, what the authors wrote in L377-379 possibly applies to irradiance effects as well.

*Thank you for these points. We now note on L373-374 that we have chosen a single spatial scale for correlative analysis, acknowledging that some predictors capture variability at different spatial scales. We have also revised Fig. 7 to include a subplot of correlations per month and have expanded the discussion throughout (L420-424 & L436-437) to discuss these relationships.*

L398: Please examine your argumentation more in depth and rephrase accordingly, because it is currently at odds with the reference chosen to support it. Sunda et al. 2002 found that Fe-induced oxidative stress upregulated DMSP synthesis and its cleavage to DMSP intracellularly. Note that DMSP cleavage can be catalyzed by lyases, or proceed through OH radical attack on DMSP intracellularly (i.e. in species lacking DMSP lyase; Spiese et al., 2015), followed by quick DMS release through cell membranes (Lavoie et al., 2018). In general, the work of Sunda et al. is used to explain enhanced DMS leakage out of algal cells upon oxidative stress, not enhanced DMS consumption.

• Spiese, Christopher E., et al. "Dimethylsulfide membrane permeability, cellular concentrations and implications for physiological functions in marine algae." Journal of Plankton Research 38.1 (2016): 41-54.

• Lavoie, Michel, et al. "Modelling dimethylsulfide diffusion in the algal external boundary layer: implications for mutualistic and signalling roles." Environmental microbiology 20.11 (2018): 4157-4169.

*We have removed this argument to focus on the abiotic pathways.*

L410: cyclonic eddies in particular, with upwards doming isopycnals at their core?

*We now specify that Bailey et al. (2008) found increased DMS-producers within anticyclonic eddies with positive sea surface height anomalies.*

L426: note that satellite PIC reflects detached coccoliths from senescent coccolithophore blooms, and may therefore be a poor predictor of DMS production during bloom initiation and eventual plateauing.

*We have added this consideration on L454-456.*

**Minor corrections**

L172: Spearman.

*This grammatical typo has been corrected.*

**Reviewer #2**

Lines 98-100 – Did the authors check for any additional data not included in the PMEL database?

*We did not, but we are unaware of any other databases for DMS.*

Paragraph starting on line 102 – Were any of these satellite derived parameters ground-truthed against in situ data during any of the cruises?

*We did not ground truth the satellite predictors to in-situ data, per se, since there is a temporal mismatch between the monthly resolution satellite predictors used here and in-situ data from individual cruises (prior to binning). However, several studies have previously assessed the accuracy of these predictors, and we now note in L127-128 that each likely has uncertainties associated with it.*

Lines 150-152 – The Nightingale et al. (2000) parameterization of k is not really appropriate for DMS. It is becoming more and more clear that the k wind speed-based parameterization for DMS should be linear (Blomquist et al., 2017; Bell et al., 2013; Zavarsky et al., 2018).

*Thank you for this suggestion. We have revised our calculations throughout using the Goddijn-Murphy et al. (2012) parameterization, which is both linear and validated against satellite data. Relevant changes can be found in L154-161 and L284-292, Figures 1,4, and 5 and Table 2. This new parameterization has only a small affect on the magnitude of the sea-air flux estimates, and doesn't change the dominant spatial patterns reconstructed by the ML methods.*

Line 224 – typo, should be ANN not AAN

*Thank you, this is now corrected.*

Section 3.4 – It seems highly likely (and I believe the authors allude to this too in the discussion starting on line 388 paragraph) that the correlations found (especially with SSH) are indirect. The real driver of DMS distributions is likely nutrients and type of microbes present. If the SSH represents eddies carrying the relevant nutrients, SSH is not really a universal parameter that can be used to describe DMS distributions everywhere. It would be good to see how that works in other regions without much eddy activity. Were phosphate and bacteria looked into? It seems that bacterial counts and types are important, but difficult to account with things like satellite data. Is it possible to compile that info from the in-situ measurements – or is that too low resolution for the techniques?

*We agree that the usefulness of SSHA as a predictor requires further analysis, and current follow-up work is focusing on applying these techniques to the Southern Ocean (another DMS hotspot) where numerous studies have noted the importance of eddies to nutrient supply and productivity.*

*Bacteria are likely very important to DMS distributions, but in-situ data are currently too sparse (both spatially and temporally) to incorporate into these models. To our knowledge, there are also no reliable algorithms for estimating bacterial counts from satellite data. However, it may be interesting in future work to apply these machine learning techniques to bacterial counts, from which the resulting product could then be used to aid in modelling DMS.*

*We did initially assess phosphate as a potential predictor, but preliminary analyses found its inclusion yielded no substantial improvement in predictive accuracy.*

Discussion (and Intro) – Why was this region chosen instead of one with more data coverage? Or why not try two different regions and compare findings? The area and number of data points (compared to the region that is mapped) seems small (i.e., Figure 4).

*The NESAP was of interest as it is a well-known hotspot for DMS with particularly high turnover between DMS and its related compounds (see, for example, Asher et al., 2017, Herr et al. 2019, and the distributions from the Lana et al. (2011) climatology). As mentioned in L73-74, this region has benefitted from increased sampling frequency over the years with the development of novel instruments, and although the region is small, the relative proportion of data coverage to the area mapped is not unreasonable when compared to global studies using these ML techniques (ex. Roshan & DeVries, 2017). Part of the motivation for this work was also to demonstrate these models are not just limited to global scales but can be successfully applied to smaller regions and still yield good results. As mentioned above, the encouraging results found here have motivated us to apply similar analyses to Southern Ocean.*

Section 4.1 (and methods section starting on line 187) – Why don't the two iron limitation proxies resemble each other at all? Also, the use of SSN is not really a unique identifier (e.g., effect of nutrients and photochemistry, as stated in the paragraph starting on line 388). How can this be practically handled when using SSN as a predicter? And if the relationships cannot be understood, why is it used?

*The more complicated NPQ-corrected fluorescence yield ($\varphi_f$) was an early attempt to remove fluorescence associated exclusively with photoinhibition effects (i.e. excess light energy dissipated through NPQ pathways), with the assumption that the remaining fraction of the total variable fluorescence better captures the response to iron limitation from space. However, more recent work (see Westberry et al. 2019) has suggested that that the simple ratio between total variable fluorescence (including NPQ pathways) and chlorophyll can be sufficient to detect a signal physiological response from space using past iron fertilization experiments as evidence.*

*There are a few reasons we feel the inclusion of SSN is warranted. First, and most simply, model performance is degraded when SSN is excluded, indicating it does have an important, albeit partly obscure, role in the DMS dynamics within this region. Additionally, since these regression models use non-linear solving equations, predictors are also more "dynamically weighted" in space/time when compared to linear regression for example, which produces a fixed coefficient for all cases. In a practical sense, this means that these models will still benefit from inclusion of predictors such as SSN, despite our limited mechanistic understanding of the underlying relationships.*

**Reviewer #3:**

DMS derived from marine phytoplankton, is very important to the marine biogenic sulfur cycle. Model simulation is useful to understand the DMS spatial and temporal distribution. In this study Machine-learning models were used to present the DMS flux and the relationship between DMS concentration and SSN, PAR, SSHA in the NESAP. The comments are as follow:

1. What is the advantage of machine-learn algorithms for the DMS simulation, comparing with other models. As machine-learning algorithms requires large datasets for the training and testing process. How to solve the problem of sparsity observation data for different ocean.

   *One of the central advantages is that these techniques can be robust to complex relationships, which can result in improved predictions relative to other models in areas with limited spatial coverage, as shown in this study. Data sparsity, however, does remain a challenge whether using traditional statistics or machine learning techniques, and machine learning may not be the best approach in some areas with little coverage. In cases such as these, using data from similar regions with known analogous oceanographic conditions may be helpful in correctly training these models. Increasing data frequency and coverage of DMS observations in the coming years will also undoubtedly help.*

2. In the manuscript, the summer time DMS data from 1997 to 2017 was used in the modeling simulation. It is obscure that the author used the average of DMS concentration from 1997 to 2017. I suggest the author presents the modeling results of the temporal distribution of DMS from 1997 to 2017.

   *We considered this approach, but found the data were still too sparse (both spatially and temporally) for the models to accurately represent inter-annual changes.*

3.  Line301-303 Why the DMS correlated well with SST?

*It is likely that, within this region, the correlation between DMS and SST mainly represents frontal patterns induced from mixing, which we allude to in the discussion with regards to SSHA patterns (L451-454).*

**Reviewer #4**

McNabb and Tortell present an improved prediction of DMS distributions in the NE Subactic Pacific which they achieve using machine learning algorithms. Understanding air-sea exchange of DMS is important for understanding the marine source of sulfate aerosols to the atmosphere, which act as cloud condensation nuclei. This work demonstrates that both of the machine learning techniques applied to this dataset (RFR and ANN) provide superior fits to the observations than were obtained using previously developed regressions. The paper is well written and the methodology and conclusions are sound. I think it is deserving of publication in Biogeosciences. However, I have some suggestions to help strengthen the manuscript.

I appreciated the analysis of impact of gridding resolution on the results. However, I wonder about the impact of binning the DMS data monthly regardless of year. Looking at the data in Figure 4, there is significant patchiness which I can only imagine is temporally and spatially variable. Given the power of the machine learning algorithms, why not use the full complexity of the dataset and pair the DMS observations with the closest (spatially and temporally) measurement of the predictor data sources?

*We considered this, but there were couple of factors that influenced our decision to use a monthly resolution. Training these models on daily resolution data could introduce bias due to autocorrelation among observations from the same cruise (L426-427, also discussed in Wang et al. 2020) and monthly-binned data allows us to reduce this source of uncertainty. In addition, daily resolutions have poor spatial coverage due to cloud cover creating gaps in the satellite's field of view. Monthly averaged predictors thus require less interpolation to match observations in space/time, allowing uncertainty in the models' training accuracy to be reduced.*

Two machine learning algorithms were used in this study but there wasn't a robust analysis of which one was better and why. Should future studies use one over the other? Does one need to try multiple methods? Such a discussion would be a valuable addition.

*We feel that the results presented largely show a strong agreement between the two methods (discussed in Sec. 3.2), as illustrated by the similar predictive accuracy (Fig. 3), spatial distributions (Fig. 4), and coherent predictor correlations (Fig. 6,7). Although there are some areas where the models deviate spatially, these are also areas with poor observational coverage (L271-273), which makes it difficult to ascertain whether one model's estimates are superior to the other. Future studies will likely benefit from applying both approaches to other regions of interest for DMS, where differences in algorithm performance may become more apparent.*

Minor comments:

- The methods are very sparse. More information on the machine learning algorithms should be included (e.g. was this done with a package? If so which one?) This is in the 'code availability' statement to some extent but should be included in the methods along with a brief description of the algorithms and differences between the two.

*We have expanded Sec. 2.2 (L138-145) to briefly describe the two algorithms used. We have also added a line noting the specific package/functions used (L153-154).*

- Only 20% of the data was held back for testing. It seems that it would be better to have a 50/50 split to provide a sufficiently large dataset for testing to confirm the robustness of the results.

*The major limitation of these machine learning algorithms is that their performance is sensitive to the size of the training dataset. As a result, the typical approach is to feed a larger fraction of the data (ex. 70%, see Weber et al. (2019); Roshan & DeVries (2017)) into the training process to allow for appropriate learning of the underlying patterns. In contrast to these global studies, we have chosen a slightly more restrictive train:test split of 80:20 to compensate for the reduced sample size and smaller geographic extent associated with a single region.*

- Are there any issue with correlations between the predictor variables? For example, many are derived from MODIS and so should have inherent correlations (ie not independent measurements).

*There is likely some inherent covariance between predictors, to a degree, given their distributions are dependent on similar processes (for example: circulation patterns, or nutrient depletion via biological production). We note, however, that we have taken steps to reduce any covariance that may confound the models results, such as including only a single biological predictor (see Sec. 2.6) and iteratively testing the addition of each new predictor on the RFR and ANN performance during development (for example, the extinction coefficient, Kd, was removed as it decreased $R^2$ due to covariance with other predictors).*

- Figure 1: It seems a bit surprising that the R2 value decreases so dramatically with resolution but the DMS flux barely changes. Is this just due to the large spatial variability in the flux?

*Yes, this is due to the spatial variability.*

Line 37 missing an 'a' —> by a suite of environmental …

*Thank you, this has been corrected.*

Line 152: typo? Should it be modified from?

*Thank you, this was a typo. In response to another reviewer's comments, this section has now been rephrased using a new k parameterization.*

Eq 4: are the coefficients provided anywhere?

*There are no coefficients for Eq. 4 (SRD) which is used within the VS07 model.*

Figure 1: caption refers to green lines/symbols instead of black

*Thank you, this has been corrected.*

Line 261: it would be helpful to provide the fractional area represented by the study region. For example, if it accounts for only <1% but accounts for 4-8% that is more impactful.

*We have removed this line in response to another reviewer's comment. In short, we have reevaluated our calculations to report the summertime regional-averaged fluxes only as Tg S, as assumptions in the conversion to an annual flux estimate were likely erroneous.*

Line 461: it should be "approach for modeling"

*Thank you, this has been corrected.*

---

## Author Response (AR2)

The authors have addressed the comments adequately.

Just a note, when I asked if the you looked for other data, I meant in actual papers and not in a database. There are many folks who do not submit their data to the PMEL database, but still publish their work.

Section 2.3 - thank you for correcting the k values. Did the you change the Sc according to ambient conditions before computing the fluxes?

*Yes, this was accounted for within the k parametrization derived by Goddijn-Murphy et al. (2012).*